# A climate data record of atmospheric moisture and sea surface temperature from satellite observations

Yixiao Fu [1], Cheng-Zhi Zou [2], Peng Zhang [3], Banghai Wu [1], Shengli Wu [4], Shi Liu [1,4], and Yu Wang [1]

[1]School of Earth and Space Sciences, CMA-USTC Laboratory of Fengyun Remote Sensing, University of Science and Technology of China, Hefei, 230026, China
[2]Independent Scholar
[3]Meteorological Observation Center, China Meteorological Administration, Beijing, 100081, China
[4]National Satellite Meteorological Center, China Meteorological Administration, Beijing, 100081, China

*Correspondence to*: Yu Wang (wangyu09@ustc.edu.cn)

**Abstract.** We developed a climate data record (CDR) of atmospheric column water vapor (CWV) and sea surface temperature (SST) under oceanic rain-free conditions using over two decades of observational records from three satellite instruments: the Advanced Microwave Scanning Radiometer for Earth Observing System (AMSR-E), the MicroWave Radiation Imager (MWRI), and the Advanced Microwave Scanning Radiometer-2 (AMSR2). The AMSR-E and AMSR2 satellites operated in near-stable orbits, while the MWRI experienced orbital drifts of nearly 1 h during its operational period. The CWV and SST products were retrieved from well-recalibrated level-1 brightness temperatures observed by common channels with the same frequencies on these instruments, designed for measuring these properties. Adjustments for diurnal drifting errors caused by orbital drift were applied to MWRI data using a semi-physical model developed in previous studies. The combination of prior recalibration and diurnal drift adjustment ensured inter-satellite consistency in the CDRs. Compared to in-situ radiosonde and buoy observations, the biases and root mean square errors of the CDRs are within 0.1 mm and 4.4 mm for CWV, and 0.2 K and 1.6 K for SST, respectively. Long-term trends of the retrieved CWV and SST align with observations from the Global Navigation Satellite System (GNSS) and the Global Tropical Moored Buoy Array (GTMBA) products. The global oceanic trends of CWV and SST were 0.39 mm decade$^{-1}$ and 0.16 K decade$^{-1}$, respectively, over the period 2002–2022. Inter-consistency between CWV and SST, as well as layer-mean temperatures derived from satellite microwave sounder observations, was examined and compared with climate model simulations from phase 6 of the Coupled Model Intercomparison Project (CMIP6). It was found that the trend ratio of the retrieved CWV to SST was 9.9% K$^{-1}$ in the tropics, which closely aligns with CMIP6 models. These validation results indicate that the presented CDR has high accuracy and is suitable for long-term climate change research. The CDR dataset is publicly available at https://doi.org/10.5281/zenodo.14539414 (Fu et al., 2024).

## 1 Introduction

Atmospheric column water vapor (CWV) and sea surface temperature (SST) are two essential climate variables (ECVs; Zemp et al., 2022). The long-term trend of SST is a key indicator of human-induced global warming (Fox-Kemper et al., 2021). Changes in SST also influence the ocean-atmosphere exchange of sensible and latent heat fluxes, as well as atmospheric dynamic and thermodynamic processes, through ocean-atmosphere coupling (Huntington et al., 2006; Wang J. H. et al., 2016; Minnett et al., 2019). Similarly, long-term changes in CWV affect the global water cycle, particularly the frequency and severity of heavy precipitation and drought events (Douville et al., 2021), which have significant impacts on human life and economies. Understanding the long-term changes in CWV and SST on a global scale is therefore of fundamental importance for climate change mitigation and adaptation efforts (Banzon et al., 2016; Ferreira et al., 2019; Blunden and Boyer, 2022).

Satellite remote sensing is the only means available to provide CWV and SST observations with global coverage. Satellite climate data records (CDRs) are essential for investigating long-term changes in CWV and SST (Mears et al., 2018; Minnett et al., 2019). A CDR is defined as a time series of measurements with sufficient length, consistency, and continuity (Council et al., 2004). Establishing a multi-decade CDR using a single instrument is challenging since most satellite instruments are designed for weather monitoring and have much shorter operational lifespans than required for climate change research. To overcome this challenge, it is necessary to splice together observations from the same or similar sensors onboard multiple satellites. Intercalibration between satellite instruments, which aims to remove inter-satellite inconsistencies, is a critical step in CDR development.

Several satellite-based CWV and SST CDRs have been developed and extensively used for climate trend studies (Blunden and Boyer, 2022; Santer et al., 2021; Schröder et al., 2016). The CWV CDRs include merged satellite products provided by Remote Sensing Systems (RSS) (Mears et al., 2018), the Hamburg Ocean Atmosphere Parameters and Fluxes from Satellite Data (HOAPS) from the European Organisation for the Exploitation of Meteorological Satellites' (EUMETSAT's) Satellite Application Facility on Climate Monitoring (CM SAF) (Andersson et al., 2010), and the National Aeronautics and Space Administration (NASA) Water Vapor Project–MEaSUREs (NVAP-M) from the NASA Making Earth Science Data Records for Use in Research Environments (MEaSUREs) program (Vonder Haar et al., 2012). For SST, satellite and in-situ observations are fused to produce global analyses, such as the Hadley Center Sea Ice and SST dataset (HadISST; Rayner et al., 2003) and National Oceanic and Atmospheric Administration's (NOAA's) Optimum Interpolation SST (OISST; Huang et al., 2020).

Despite extensive studies on the long-term trends in individual CWV and SST products, large uncertainties remain when their trends are analyzed for climate covariability (Wang J. H. et al., 2016; Mears et al., 2018; Santer et al., 2021). The Clausius-Clapeyron equation predicts a trend ratio of approximately 7% $K^{-1}$ for CWV relative to SST over the tropical oceans (Wentz and Schabel, 2000; Held and Soden, 2006; Trenberth et al., 2005; Wang J. H. et al., 2016), assuming constant relative humidity. This predicted ratio is well-replicated in simulations by phases 5 and 6 of the Coupled Model Intercomparison Project (CMIP5 and CMIP6) models (Santer et al., 2021). Such results provide a strong constraint on the observational trend ratio of CWV

relative to SST (Santer et al., 2021). However, observational datasets often show CWV-to-SST ratios and other atmospheric temperature trends that deviate significantly from theoretical expectations and climate model simulations (Santer et al., 2005; Mears et al., 2007; Santer et al., 2021). It remains unclear whether these differences are caused by trend errors in observed CWV, SST, or both (Santer et al., 2021). This highlights the need for improved observational datasets of CWV and SST and a deeper understanding of the error sources associated with their CDRs.

Two main sources of error arise when developing a satellite-based CDR. First, measurement differences between sensors can result from variations in preprocessing, calibration approaches, and hardware designs, such as channel frequency, Earth incident angle, and bandwidth (Zou et al., 2018; Wu et al., 2020; Liu et al., 2023). Second, different satellites often have different local observation times depending on their orbits. Even within the same satellite, local observation times may vary over time due to orbital drift (Zou et al., 2018; Bojanowski and Musial, 2020; Lang et al., 2020). For physical variables with significant diurnal cycles, these variations can lead to large measurement differences (O'Dell et al., 2018), potentially introducing spurious trend signals in CDRs if not corrected. Different algorithms for correcting diurnal drift effects can also lead to trend differences in CDRs (Mears and Wentz, 2016; Po-Chedley et al., 2015; Zou et al., 2023).

We aim to develop a new set of CWV and SST CDRs, both using observations from three satellite instruments with close local observation time: the Advanced Microwave Scanning Radiometer for Earth Observing System (AMSR-E) onboard NASA's Aqua satellite (Kawanishi et al., 2003), the Advanced Microwave Scanning Radiometer-2 (AMSR2) onboard JAXA's Global Change Observation Mission first-Water (GCOM-W1) satellite (Maeda et al., 2016), and the MicroWave Radiometer Imager (MWRI) onboard the FengYun-3B (FY3B) satellite (Yang et al., 2011), developed by the National Satellite Meteorological Center of the China Meteorological Administration (NSMC). These instruments share the same channel frequencies, in the microwave range from 10.65 to 89 GHz, designed to measure water-vapor-related geophysical parameters with both vertical and horizontal polarizations. In particular, the MWRI can serve as an effective bridging instrument to fill the observation gap of approximately nine months, from October 2011 to July 2012, between AMSR-E and AMSR2, making continuous and complete CDRs possible for the period from 2002 to 2022. Recently, Wu et al. (2020) recalibrated the level-1 brightness temperatures (TBs) for these three sensors and constructed a consistent fundamental CDR (FCDR) for the last two decades using several intercalibration approaches. Additionally, it is worth noting that two of the three satellites have stable local overpass times, while one experienced orbital drift, as discussed below. Previous studies (Zou et al., 2018, 2021) demonstrated that satellite observations in stable sun-synchronous orbits produce highly accurate CDR trends. For satellites with orbital drift, Zou et al. (2023) developed a novel semi-physical model to effectively remove diurnal drifting errors in satellite microwave observations. These new developments have provided a solid foundation for constructing more accurate CWV and SST CDRs. Here, we apply these recent advances to develop CWV and SST CDRs from satellite microwave observations. Our results demonstrate that the CWV and SST CDRs from the same observations produced in this study exhibit trend ratios consistent with climate model simulations and theoretical expectations. The rest of this article is organized as follows. Section 2 presents the datasets used for validating the retrieved CDRs. Section 3 describes the source FCDR and ECV algorithm. Section 4 performs the validation of the CWV and SST products against various observations from other instruments and climate

reanalyses. Section 5 examines the ratios of CWV to SST and satellite microwave sounder-based atmospheric layer temperatures over the tropical ocean. Finally, conclusions and discussions are provided in Sect. 7.

## 2 Validation data

### 2.1 Radiosonde and Buoy Observations

In-situ measurements are widely used to validate retrievals from satellite observations at the pixel level (Gentemann and Hilburn, 2015; Wentz and Meissner, 2000). In this study, radiosonde observations (RAOBs) from the Integrated Global Radiosonde Archive (Durre et al., 2006) are used to evaluate the retrieved CWV ($CWV_{RTV}$). Since the retrieved CWV is available only over the ocean, RAOB sites on 63 islands are selected for validation (Fig. 1). The equivalent CWV from RAOB ($CWV_{RAOB}$) is calculated from RAOB profiles, which include pressure, air temperature, and dew point depression (Alishouse et al., 1990). The RAOB data has undergone a multi-stage quality assurance process, including persistence checks, climatological outlier removal, and vertical/temporal consistency tests, which ensure internal coherence and minimize undetected errors to about 1.1% (Durre et al., 2008). During validation, only $CWV_{RTV}$ pixels closest to $CWV_{RAOB}$ are selected for comparison. The collocation criteria for $CWV_{RTV}$ and $CWV_{RAOB}$ are a spatial distance of 60 km and a temporal interval of 3 h (Wang et al., 2009).

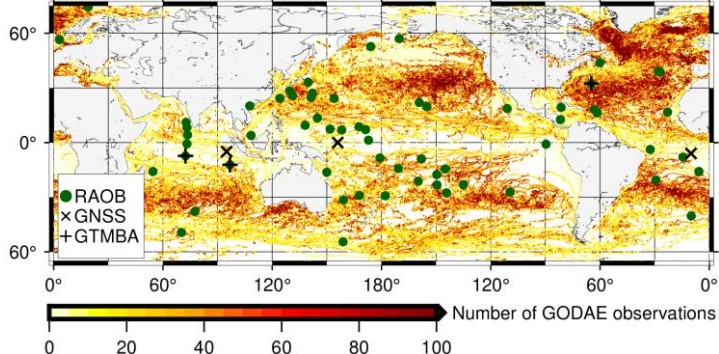

**Figure 1: Geographic locations of RAOB, GNSS, and GTMBA sites, along with GODAE measurements used for validation in this study. The number of GODAE buoy observations in 2021 is shown within 0.25° grid boxes.**

The U.S. Global Ocean Data Assimilation Experiment (GODAE) from the Fleet Numerical Meteorology and Oceanography Center (FNMOC) has collected global SST observations ($SST_{GODAE}$) from ships, moored buoys, and drifting buoys (Fig. 1). These observations are used to validate the retrieved SST ($SST_{RTV}$) on a pixel-by-pixel basis. GODAE files include time, latitude, longitude, SST, the probability of gross error—assuming a normal probability density function for SST errors (Cummings, 2011)—and other metadata. The probability of gross error is used to guarantee data quality and consistency, ensuring that only high-confidence data (defined as those with a probability less than 0.6 K) are used here (Gentemann and Hilburn, 2015). For validation, only the satellite observation pixel closest to each in-situ measurement is selected. The high spatial and temporal resolutions of GODAE observations enable a stringent collocation criterion of 0.1° and 6 min.

In addition to GODAE, the Global Tropical Moored Buoy Array (GTMBA) provides systematic and sustained SST observations for monitoring tropical atmospheric and oceanic interactions, supporting climate research and assessment (McPhaden et al., 2023). GTMBA consists of three moored buoy arrays: the Tropical Atmosphere Ocean/Triangle Trans-Ocean Buoy Network (TAO/TRITON), the Prediction and Research Moored Array in the Atlantic (PIRATA), and the Research Moored Array for African-Asian-Australian Monsoon Analysis and Prediction (RAMA) (Foltz et al., 2019; Smith et al., 2019; Beal et al., 2020). These buoy arrays are located in the tropical waters of the Pacific (0° N, 156° E), Atlantic (6° S, 10° W), and Indian Oceans (5° S, 95° E) (Fig. 1). Observations from these buoys provide the longest continuous time series for validating the long-term trends of $SST_{RTV}$.

## 2.2 Global Navigation Satellite System (GNSS) CWV Retrievals

The GNSS deployed at ground stations provides moisture profile retrievals (Bock, 2020). GNSS retrievals offer advantages such as long-term stability, high accuracy, and high temporal resolution. They are often used for monitoring climate change and validating satellite and reanalysis products (Mears et al., 2018; Chen et al., 2021; Yuan et al., 2023; Blunden and Boyer, 2022). For comparison, the GNSS moisture profiles are converted to equivalent CWV. Monthly mean data from three isolated island stations with long observation records—BRMU (32.37° N, 64.70° W), COCO (12.19° S, 96.83° E), and DGAR (7.27° S, 72.37° E) (Fig. 1)—are selected to validate the long-term trend in $CWV_{RTV}$.

## 2.3 SST analysis product

The NOAA OISST (also known as Reynolds' SST) is a long-term, globally gridded SST analysis product (Huang et al., 2020). It is developed by optimally integrating SST retrievals from satellite Advanced Very High Resolution Radiometer (AVHRR) observations with in-situ buoy and ship measurements. This SST dataset has been widely used in trend studies (Blunden and Boyer, 2022; Banzon et al., 2016). In this study, the monthly mean OISST product (version 2.1) with a horizontal resolution of 0.25° is used for climate trend comparisons with our $SST_{RTV}$.

## 2.4 RSS CDRs

Among the CWV CDRs, the RSS product (available at http://www.remss.com) has the longest time period with documented accuracy (Mears et al., 2018). This CWV product is a monthly gridded dataset with 1° resolution, created by averaging observations from 11 inter-calibrated satellites spanning 1987 to the present (Mears et al., 2018). In this study, the RSS CWV ($CWV_{RSS}$) product from 2002 to 2022 is used for climate trend comparisons with $CWV_{RTV}$. However, the RSS SST cannot be retrieved for several of these 11 satellites (e.g., F08–F18) due to the absence of low-frequency (10 GHz) channels on some of the passive microwave (PMW) imagers. To address this, another RSS product, named the Air-Sea ECV Merged Microwave CDR (hereinafter referred to as RSS-CDR), was developed by the RSS group (Wentz and the RSS team, 2021) using RSS CWV and OISST instead of RSS SST. This later RSS-CDR product, available on a monthly global 2.5°×2.5° grid from July 1987 through February 2021, is used for validation purposes in this study.

## 2.5 Reanalysis Datasets

Reanalysis data provide long-term, globally continuous ECVs within the Earth-atmosphere system. These datasets are generated by assimilating various observed data into model simulations using a fixed atmospheric numerical weather forecast model over time (Dee et al., 2011; Kalnay et al., 1996). Since different observations vary in accuracy and spatiotemporal coverage, optimal algorithms are required to blend observations with model simulations, ensuring that reanalysis data remain consistent with both physical principles and observational constraints.

In this study, we use the fifth generation of the European Centre for Medium-Range Weather Forecasts (ECMWF) atmospheric reanalysis products (ERA5; Hersbach et al., 2020) for climatological comparisons with our retrieved CWV and SST. ERA5 is produced with an advanced atmospheric model and data assimilation system, offering improved accuracy and resolution compared to its predecessor, ECMWF ERA-Interim. Monthly mean ERA5 products with a 0.25°×0.25° resolution for CWV ($CWV_{ERA}$) and SST ($SST_{ERA}$) are utilized to compare the climate trends of our satellite-retrieved CDR and SST over the same time period.

## 2.6 CMIP6 product

The CMIP6 (Eyring et al., 2016) provides multi-decadal simulations of various atmospheric parameters, including atmospheric moisture profiles and SST, from multiple ocean-atmosphere coupled global climate models. In these simulations, the thermodynamic properties of the climate system are controlled and constrained by interior physical mechanisms (Santer et al., 2021). One such process is that the tropical atmosphere (20˚S–20˚N) generally follows the Clausius-Clapeyron law due to adiabatic moisture processes (Trenberth et al., 2005; Wang J. H. et al., 2016). As a result, the trend ratio of CWV over temperature remains relatively constant across different climate model simulations, although individual trends of CWV and temperature vary among models due to differences in climate sensitivities and the coupling of dynamical and radiative processes. This characteristic can be used to constrain satellite-observed trend ratios (Santer et al., 2021; Wang J. H. et al., 2016), including the ratio of CWV over SST (denoted as $R_{\{CWV/SST\}}$), the ratio of CWV over mid-tropospheric temperatures (TMT; denoted as $R_{\{CWV/TMT\}}$ ) and the ratio of CWV over lower-tropospheric temperate (TLT; denoted as $R_{\{CWV/TLT\}}$).

CMIP6 includes simulation results from multiple experiments designed to address specific scientific questions (Eyring et al., 2016). Among these, the historical simulation experiment covers the climate change process from 1850 to 2014, while the Scenario Model Intercomparison Project (ScenarioMIP) provides projections of climate change beyond 2015 based on different Shared Socioeconomic Pathway (SSP) scenarios. For example, SSP585 is a climate prediction under extreme emission scenarios, which can be spliced with historical simulations to obtain continuous simulation results (Santer et al., 2021). In this study, outputs from 28 spliced models in CMIP6 (Table 1) during 2003–2020 from the tropics were selected to validate the covariation trend of the current CDR ECVs. To compare with satellite observations, the air temperature profiles in CMIP6 models are converted to the TMT and the TLT using their weighting functions (Santer et al., 2021).

**Table 1: Basic information of CMIP6 models. The term "Ensemble Member" refers to the initial conditions of the climate model and the identifiers of the different parameters. For example, r1i1p1f1 indicates the simulation result for the first initial condition, the first physical parameterization setting, and the first external forcing setting, while r1i1p1f2 is the same as r1i1p1f1 except for the second external forcing.**

| Num. | Forecast Centre | Model | Ensemble Member |
|---|---|---|---|
| 1 | Australian Community Climate and Earth System Simulator (ACCESS), | ACCESS-CM2 | r1i1p1f1 |
| 2 | Australia | ACCESS-ESM1-5 | r1i1p1f1 |
| 3 | Alfred Wegener Institute, Helmholtz Centre for Polar and Marine Research, Germany | AWI-CM-1-1-MR | r1i1p1f1 |
| 4 | Beijing Climate Center, China Meteorological Administration, China | BCC-CSM2-MR | r1i1p1f1 |
| 5 | Chinese Academy of Meteorological Sciences, China | CAMS-CSM1-0 | r1i1p1f1 |
| 6 | Community Earth System Model, National Center for Atmospheric Research (NCAR), USA | CESM2-WACCM | r1i1p1f1 |
| 7 | Centre National de Recherches Météorologiques, France | CNRM-CM6-1-HR | r1i1p1f2 |
| 8 | | CNRM-ESM2-1 | r1i1p1f2 |
| 9 | Energy Exascale Earth System Model, U.S. Department of Energy, USA | E3SM-1-0 | r1i1p1f1 |
| 10 | | E3SM-1-1 | r1i1p1f1 |
| 11 | | E3SM-1-1-ECA | r1i1p1f1 |
| 12 | EC-Earth consortium, Europe | EC-Earth3 | r1i1p1f1 |
| 13 | | EC-Earth3-CC | r1i1p1f1 |
| 14 | | EC-Earth3-Veg | r1i1p1f1 |
| 15 | | EC-Earth3-Veg-LR | r1i1p1f1 |
| 16 | Flexible Global Ocean-Atmosphere-Land System Model, Institute of Atmospheric Physics, Chinese Academy of Sciences, China | FGOALS-g3 | r1i1p1f1 |
| 17 | Geophysical Fluid Dynamics Laboratory, NOAA, USA | GFDL-CM4 | r1i1p1f1 |
| 18 | Institute of Numerical Mathematics, Russian Academy of Sciences, Russia | INM-CM4-8 | r1i1p1f1 |
| 19 | | INM-CM5-0 | r1i1p1f1 |
| 20 | Korea Institute of Atmospheric Prediction Systems (KIAPS), Korea | KACE-1-0-G | r1i1p1f1 |
| 21 | Korea Institute of Ocean Science and Technology, Korea | KIOST-ESM | r1i1p1f1 |
| 22 | Model for Interdisciplinary Research on Climate, Japan | MIROC6 | r1i1p1f1 |
| 23 | | MIROC-ES2L | r1i1p1f2 |
| 24 | Max Planck Institute for Meteorology, Germany | MPI-ESM1-2-HR | r1i1p1f1 |
| 25 | | MPI-ESM1-2-LR | r1i1p1f1 |
| 26 | Meteorological Research Institute, Japan | MRI-ESM2-0 | r1i1p1f1 |
| 27 | Norwegian Climate Centre, Norway | NorESM2-LM | r1i1p1f1 |
| 28 | | NorESM2-MM | r1i1p1f1 |

## 2.7 Satellite temperature data

Three TMT and TLT products derived from satellite microwave sounder observations—NOAA's Center for Satellite Applications and Research (STAR; version 5.0; Zou et al., 2021, 2023), RSS (version 4.0; Mears and Wentz, 2016, 2017), and the University of Alabama in Huntsville (UAH; version 6.0; Spencer et al., 2017)—are used to examine the covariance of $CWV_{RTV}$ with observed tropospheric temperatures in this study. STAR, RSS, and UAH are all monthly mean datasets on a global $2.5° \times 2.5°$ grid. Their temperature trends over the tropical ocean from 2003 to 2020 were used to construct observed

trend ratios of CWV over tropospheric temperatures and to compare them with climate model results.

It is worth noting that STAR has constructed a reference TMT and TLT dataset based solely on microwave sounder observations in stable orbits during the period from 2002 to the present (Zou et al., 2021). This reference time series achieves an accuracy of 0.012 K decade$^{-1}$ in trend detection (Zou et al., 2021). The STAR V5.0 dataset maintains this trend detection accuracy during 2002–present by inter-calibrating satellites with orbital drifts to the reference dataset (Zou et al., 2023). The

high trend accuracy in the STAR dataset serves as a robust reference for evaluating the trend accuracy of $CWV_{RTV}$ presented in this study. This is demonstrated in Sect. 5, where we examine the ratios of CWV over tropospheric temperatures.

## 2.8 Validation Framework of Reference Datasets

To ensure a structured evaluation of the CDR quality, we systematically validated the retrieved CWV and SST products across multiple dimensions: retrieval accuracy, local and global long-term trends, and climate trend covariability. The various datasets

mentioned above differ in spatial coverage, temporal extent, measurement principles, and are therefore suitable for different validation tasks. Table 2 summarizes the validation objectives, evaluation metrics, corresponding reference datasets, and their independence from the retrievals.

**Table 2: Reference datasets used in validation. (RMSE: Root mean square error; CC: Correlation coefficient)**

| Validation Objectives | Variable | Evaluation Metric | Reference Dataset | Dependency |
|---|---|---|---|---|
| **Retrieval accuracy** | CWV | Bias, RMSE, CC | RAOB | Fully independent |
| | SST | Bias, RMSE, CC | GODAE | Fully independent |
| **Regional variability and trend** | CWV | Trend, CC | GNSS | Fully independent |
| | SST | Trend, CC | GTMBA | Fully independent |
| **Global variability and trend** | CWV | Trend, CC | RSS | AMSR-E and AMSR2 observations included |
| | SST | Trend, CC | OISST | Fully independent |
| | CWV and SST | Trend, CC | ERA5 | AMSR-E and AMSR2 observations included |

| | CWV and SST | Trend, CC | RSS-CDR | AMSR-E and AMSR2 observations included |
|---|---|---|---|---|
| **Climate trend covariability** | CWV, SST, TLT and TMT | $R_{\{CWV/SST\}}$, $R_{\{CWV/TLT\}}$ and $R_{\{CWV/TMT\}}$ | CMIP6 | Fully independent |
| | TLT and TMT | $R_{\{CWV/TLT\}}$ and $R_{\{CWV/TMT\}}$ | STAR | Fully independent |
| | TLT and TMT | $R_{\{CWV/TLT\}}$ and $R_{\{CWV/TMT\}}$ | RSS | Fully independent |
| | TLT and TMT | $R_{\{CWV/TLT\}}$ and $R_{\{CWV/TMT\}}$ | UAH | Fully independent |

## 3 CDR Development for CWV and SST


The development of CWV and SST CDRs requires well-intercalibrated and recalibrated radiance or TB data records at the satellite swath pixel level (level-1). The continuous and consistent radiance dataset, consisting of recalibrated observations from different sensors, is referred to as the FCDR (Liu et al., 2023; Poli et al., 2023). Recently, Wu et al. (2020) developed a continuous FCDR for the past two decades by applying several intercalibration approaches to recalibrate the observed TBs of the three sensors: AMSR-E, MWRI, and AMSR2. By examining long-term changes in these recalibrated TBs over the global

ocean, it is demonstrated that this FCDR is consistent and homogeneous enough to be used for obtaining CDRs of water cycle-related variables for climate research. Accordingly, this TB FCDR for the three sensors–AMSR-E, MWRI and AMSR2–is used for our CWV and SST retrievals, which includes TBs from both vertically (V) and horizontally (H) polarized channels at five common frequencies: 10.65, 18.7, 23.8, 36.5, and 89.0 GHz (hereinafter referred to as 10V/H, 18V/H, 23V/H, 36V/H, and 89V/H for convenience). The observations from the three sensors spanned 20 years, from June 2002 to May 2022 (Fig. 2).

Among them, the AMSR-E and AMSR2 satellites maintained near-stable orbits throughout this period, with their ascending Local Equator Crossing Times (LECTs) around 1:30 PM. In contrast, the MWRI satellite's orbit drifted from 1:35 PM in 2011 to 3:40 PM in 2020 (Fig. 2).

The AMSR-E and AMSR2 FCDRs covered the periods from June 2002 to August 2011 and July 2012 to May 2022,

respectively. To ensure continuity and stability of the CDR, observations from MWRI during June 2011 to April 2015 were selected to bridge the temporal gap between AMSR-E and AMSR2. This selection of FY3B data allows a few months of overlap with AMSR-E (June to September 2011) and more than two years of overlap with AMSR2, facilitating inter-satellite calibration. It is important to note that orbit changes in May 2015 may have introduced bias jumps, so MWRI data after this time were excluded.

In the FCDR development by Wu et al. (2020), AMSR2 was used as a reference to recalibrate AMSR-E and MWRI, eliminating hardware differences between the sensors (e.g., those due to Earth incidence angle variations) using principal

component analysis. This unique advantage of consistent TBs between various sensors at the observation level enables the FCDR to be directly utilized for CDR retrievals.

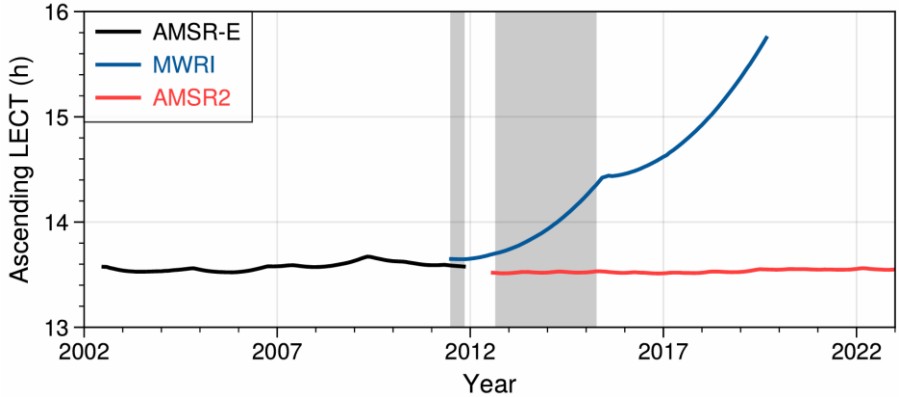

**Figure 2: Ascending Local Equator Crossing Times (LECTs) for the three satellites—Aqua, FY3B, and GCOM-W1—used in CWV and SST retrievals. The grey areas represent the overlap periods of MWRI with AMSR-E or AMSR2.**

Table 3 summarizes the instruments and their time coverage used to construct the current CDR. To ensure consistency and long-term stability of the retrieved climate variables, a series of processing steps and adjustments were implemented. These steps include: i) removing precipitation and sea ice pixels, ii) correcting for diurnal drifts in MWRI, iii) retrieving CWV and SST, iv) spatially gridding the data onto a global 0.25° × 0.25° grid, v) merging of multi-sensor anomalies into a continuous time series, and vi) comprehensive uncertainty quantification at each processing level. Details of each step are provided below.

**Table 3: Satellite instruments contributing to the CDR.**

| Satellite | Instrument | Start date | End date |
|-----------|------------|------------|----------|
| Aqua | AMSR-E | 2002-06 | 2011-06 |
| FY3B | MWRI | 2011-03 | 2015-04 |
| GCOM-W1 | AMSR2 | 2012-09 | 2022-05 |

The influence of precipitation and sea ice on the CWV and SST retrievals was eliminated. A TB threshold method for precipitation identification (Durre et al., 2006) was applied to remove precipitation-affected pixels. The Moderate Resolution Imaging Spectroradiometer (MODIS) onboard the Aqua satellite provides daily gridded data on sea ice extent [MYD29; Wentz, 1997, distributed by the U.S. National Snow and Ice Data Center (NSIDC)]. This data was used to exclude pixels with sea ice. Moreover, Orbital drift caused MWRI scene TBs to develop diurnal biases due to changes in diurnal sampling over time. As a result, a diurnal drift correction was applied in this study to all channels before they were used for CWV and SST retrievals, by adjusting observations taken at different times to a common local time consistent with AMSR2, thereby empirically correcting diurnal drift biases. This process requires knowledge of diurnal cycles, which are functions of time and geolocation. Recently, Zou et al. (2023) developed a novel semi-physical model that effectively resolves diurnal cycles in satellite observations and removes diurnal drifting errors globally. In this study, we followed the procedures and diurnal correction

equations developed by Zou et al. (2023) to resolve the diurnal drifting biases in MWRI. A detailed description of the semi-physical model is referred to Zou et al. (2023). For a single satellite such as MWRI, this semi-physical model is expressed as:

$$D_{MWRI}(X, m, L) = \alpha(X) + \beta(X, m)\sin(2\pi L/24) + \gamma(X, m)\cos(2\pi L/24), \tag{1}$$

where $D_{MWRI}$ is the diurnal anomaly of MWRI at geolocation $X$, month $m$ and local time $L$; $\beta(X, m)$ and $\gamma(X, m)$ are the amplitudes of the monthly diurnal components at different geolocations. The coefficient $\alpha(X)$ is a constant changing with geolocation. The coefficients $\beta$ and $\gamma$ varies with season and are assumed as:

$$\beta = \beta_0 + \beta_1\sin(2\pi m/12) + \beta_2\cos(2\pi m/12), \tag{2}$$

$$\gamma = \gamma_0 + \gamma_1\sin(2\pi m/12) + \gamma_2\cos(2\pi m/12), \tag{3}$$

where $\beta_0$, $\beta_1$, $\beta_2$, $\gamma_0$, $\gamma_1$ and $\gamma_2$ are all constants.

The TBs after diurnal adjustment ($TB'_{MWRI}$) can be expressed as:

$$TB'_{MWRI}(X, t, m) = TB_{MWRI}(X, t, m, L) - D_{MWRI}(X, m, L), \tag{4}$$

During the overlapping period between MWRI and the other two AMSR instruments (Fig. 2), the TB differences can be expressed as:

$$TB'_{MWRI}(X, t, m) - TB_{AMSR}(X, t, m) = TB_{MWRI}(X, t, m, L) - TB_{AMSR}(X, t, m, L) - D_{MWRI}(X, m, L), \tag{5}$$

By minimizing inter-satellite differences between satellite pairs ($\Delta TB' = TB'_{MWRI}(X, t, m) - TB_{AMSR}(X, t, m)$), the coefficients $\alpha$, $\beta$ and $\gamma$ can be solved through multiple linear regression based on the monthly gridded (2.5° resolution) TBs. It is essential to note that the "semi-physical" nature of the diurnal model requires that Eqs. (1)–(5) be resolved separately for daytime and nighttime, or for ascending and descending nodes (Zou et al., 2023). This distinction arises because daytime diurnal cycles exhibit a physically-based, solar heating-induced quasi-sinusoidal pattern, whereas nighttime diurnal cycles follow a thermal decay process. In the latter case, Eq. (1) serves as an empirical approximation of the thermal decay process (Zou et al., 2023).

Figure 3 shows the global ocean mean inter-satellite difference time series between MWRI and the other two satellites, compared with their resolved diurnal cycle differences. The agreement in the temporal patterns is generally impressive across nearly all channels. Subtracting the diurnal differences yields the diurnally adjusted MWRI TBs, which are used for CWV and SST retrievals. These diurnally adjusted MWRI TBs are consistent with AMSR-E and AMSR2, as their mean biases are zero with negligible standard deviations during their overlaps. The impact of the diurnal adjustment on CWV and SST retrievals is shown below.

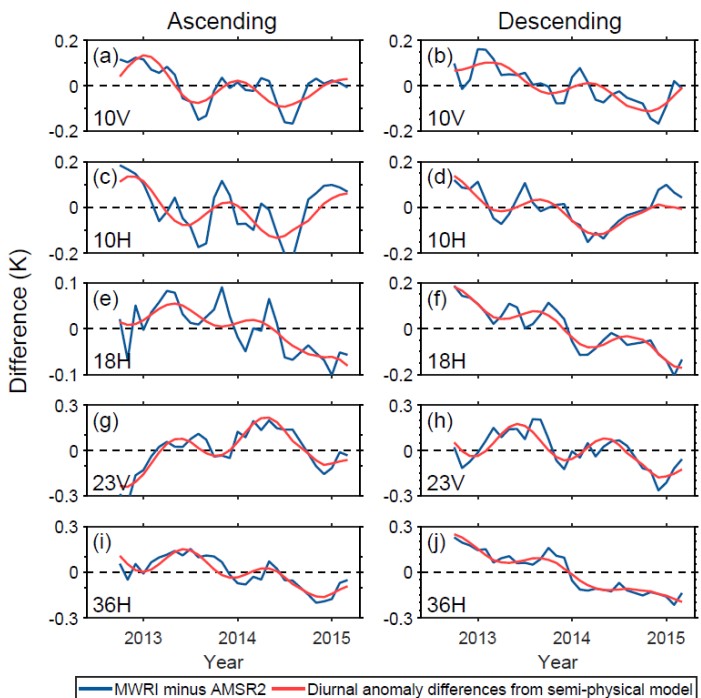

**Figure 3: Inter-satellite differences ($TB_{MWRI}(X, t, m) - TB_{AMSR}(X, t, m)$) before diurnal drift adjustment (blue lines) and diurnal anomaly differences ($D_{MWRI}(X, m, L)$, red lines) derived from the semi-physical model over the global ocean. The differences are grouped into (left) ascending and (right) descending data separately, with the channels shown from (top) to (bottom): 10V, 10H, 19H, 23V, and 37H.**

The retrieval algorithm for CWV is based on the one developed by Wang et al. (2009) for measurements from the Tropical Rainfall Measuring Mission's (TRMM) Microwave Imager (TMI). This algorithm uses TBs on five channels in the FCDR (10V, 10H, 19H, 23V, and 37H) with a logarithm relationship between the channel signals and $CWV_{RTV}$, expressed as:

$$CWV_{RTV} = a_0 + \sum_{i=1}^{5} a_i ln \left( \frac{288 - TB_i}{288} \right), \tag{6}$$

where the coefficients $a_i (i = 0, 1, \dots 5)$ in Eq. (6) were derived from multiple linear regression of simulated AMSR2 TBs from a microwave radiative transfer model (Liu, 1998, 2004) with varying CWVs and cloud parameters as inputs. These coefficients and the sensitivities of the retrieved CWV to TB for each channel are shown in Table 4. As expected, the 23V channel, near the water vapor absorption line, has the largest influence on the CWV retrievals. The advantage of this algorithm is that the retrieved CWV depends only on TBs from the five channels, without additional ancillary data. As a result, the $CWV_{RTV}$ trend is only related to the TB trends.

**Table 4: Regression coefficients and the corresponding sensitivities of CWV and SST to TBs for different PMW channels used in the retrieval algorithms. For SST, the regression coefficients represent the initial estimate.**

|  |  | Offset | TB 10V | TB 10H | TB 19H | TB 23V | TB 37H |
|---|---|---|---|---|---|---|---|
| **CWV** | $a_i$ | -9.953 | 42.890 | 29.689 | -84.776 | -44.691 | 41.433 |
| **(mm)** | Δ CWV/Δ TB |  | -0.387 | -0.151 | 0.524 | 0.732 | 0.319 |
| **SST** | $b_i$ | -559.777 | -940.886 | -406.252 | 201.324 | -72.084 | 47.956 |
| **(K)** | $c_i$ |  | -393.786 | -637.333 | 188.989 | -20.560 | 13.612 |
|  | Δ SST/Δ TB |  | 1.764 | -0.313 | 0.082 | 0.118 | -0.203 |

300

According to Wang et al. (2009), the retrieval framework for CWV can also be used to retrieve SST or surface wind speed (SWS). To alleviate SST biases in the SST retrievals associated with the assumption of a logarithm relationship between SST (SWS) and the TB [see Eqs. (2) and (3) in Wang et al. (2009) for details], a second-order term of the logarithm of TB is added in Eq. (6) as a bias correction term. This gives an $SST_{RTV}$ or $SWS_{RTV}$ retrieval algorithm expressed as:

305
$$SST_{RTV} = b_0 + \sum_{i=1}^{5} \left[ b_i \ln \left( \frac{288-TB_i}{288} \right) + c_i \ln \left( \frac{288-TB_i}{288} \right)^2 \right], \tag{7}$$

$$SWS_{RTV} = d_0 + \sum_{i=1}^{5} \left[ d_i \ln \left( \frac{288-TB_i}{288} \right) + e_i \ln \left( \frac{288-TB_i}{288} \right)^2 \right], \tag{8}$$

where the coefficients $b_i$, $c_i$, $d_i$, and $e_i$ (i = 0, 1, … 5) are derived from similar simulations as described for CWV. Equations (7) and (8) provide a reasonably good initial estimate of SST and SWS. However, the simple forms given by Eqs. (7) and (8) are incapable of fully representing the non-linearities in the relationship between TBs and SST (SWS) (Wentz and Meissner, 2007;

310 Li and Jiang, 2024). Hence, in this study, the localized algorithm proposed by Wentz and Meissner (2007) is applied to 34 SST reference values from 272 to 308 K and 34 SWS reference values from 0 to 35 m s$^{-1}$. For each reference value, the coefficients are calculated within the ranges of ±1 K for SST and ±1 m s$^{-1}$ for SWS, respectively. The final retrieval is a bilinear interpolation of the four nearest first-step retrievals in the 2-D space of SST and SWS. After retrieval, the satellite orbit data are averaged to monthly global 0.25°×0.25° gridded data for each satellite.

315 From the perspective of instrument design and microwave radiative transfer, different ECVs respond to measurements of different frequency channels in distinct ways. As shown in Table 4, low-frequency channels such as 10V and 10H are primarily sensitive to SST, while higher-frequency channels like 19H, 23V, and 37H are more responsive to CWV. As a result, although CWV and SST are both retrieved from the same set of satellite TBs, the differences in channel sensitivity and retrieval algorithm expressions result in them being physically and mathematically independent products.

320 Figure 4 shows the impact of diurnal adjustment on the consistency of the retrieved CWV and SST between different satellites. Before the adjustment, the mean biases between MWRI and the other two satellites were -0.036 mm and 0.180 K for CWV and SST, respectively, with standard deviations of 0.114 mm and 0.076 K. After the adjustment, the mean biases were reduced to 0.003 mm and 0.003 K, respectively, with standard deviations of 0.058 mm and 0.050 K.

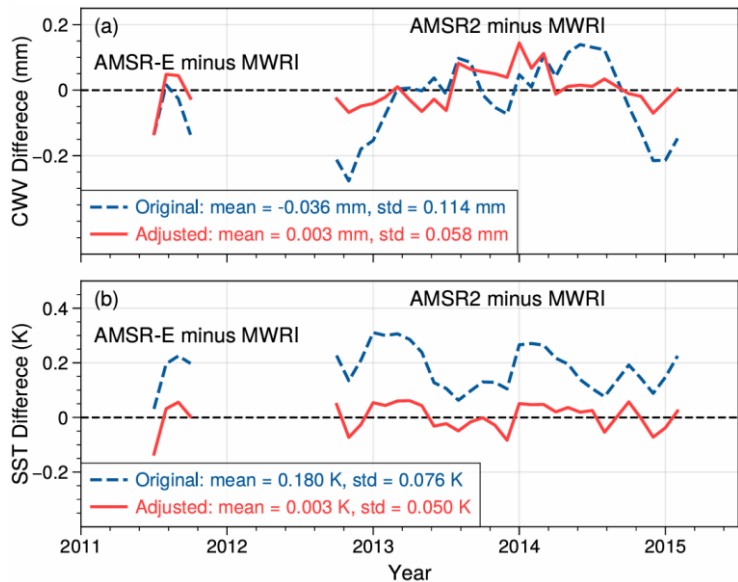

**Fig. 4 Monthly mean differences between MWRI and the other two instruments for (a) CWV and (b) SST over the global ocean during their overlapping period. The blue dashed and red solid lines indicate the differences before and after diurnal adjustment, respectively. The mean biases and standard deviations are shown in the legends.**

After diurnal adjustment, the MWRI observations were converted to resemble observations from satellites in stable orbits. As a final step to merge data from different satellites, we followed the procedure outlined by Zou et al. (2021) for merging satellites in stable orbits. The procedure is as follows: i) Calculate anomalies for ascending and descending data separately, using a monthly climatology defined for the entire observation period for each satellite. ii) Average the ascending and descending anomalies to construct daily mean anomalies. iii) Make an adjustment so that the SST or CWV anomalies of individual satellites are defined relative to the same monthly climatology. We use the AMSR2 monthly climatology as a reference and adjust the MWRI anomalies by subtracting the "monthly climatology" of anomaly differences relative to AMSR2 during their overlapping periods (Zou et al., 2021). The AMSR-E data was further adjusted to match the MWRI anomalies using their overlapping observations. After these adjustments, the anomalies from the three satellites are averaged together to generate a CDR for the entire 2002–2022 period for trend investigation.

Uncertainties in our dataset were systematically assessed across level-1 to level-3. Table 5 gives uncertainty estimates for level-1 and level-2. For level-1, the FCDR relies on pre-processed TBs from AMSR-E, AMSR2, and FY-3 MWRI. Channel-specific TB uncertainties for AMSR-E and AMSR2 are defined by the noise-equivalent temperature difference (NE$\Delta$T) and adjusted by noise amplification factors derived from the Backus–Gilbert (BG) resampling method (Kawanishi et al., 2003; Maeda et al., 2016). For MWRI, the complex recalibration introduces strong nonlinearity (Wu et al., 2020), for which analytical Jacobians are not available to uncertainty propagation. Thus, a Monte Carlo method (Merchant et al., 2017; Roebeling et al., 2025) was used to estimate the uncertainty after correction by perturbing input TBs with NE$\Delta$T and calculating the standard deviation of the corrected outputs.

**Table 5: Uncertainty Estimates for level-1 TBs and level-2 retrievals.**

| Product Level | Channel / Parameter | | AMSR-E | MWRI | AMSR2 |
|---|---|---|---|---|---|
| **level-1** | TB uncertainty (K) | 10V | 0.29 | 0.26 | 0.28 |
| | | 10H | 0.29 | 0.25 | 0.28 |
| | | 18H | 0.13 | 0.18 | 0.18 |
| | | 23V | 0.12 | 0.18 | 0.15 |
| | | 36H | 0.12 | 0.19 | 0.15 |
| **level-2** | Retrieval uncertainty | CWV (mm) | 0.54 | 0.55 | 0.55 |
| | | SST (K) | 0.55 | 0.50 | 0.53 |

For level-2, total retrieval uncertainty includes both the propagation of TB uncertainty and the intrinsic retrieval algorithm error. The CWV algorithm allows both Jacobian-based and Monte Carlo-based propagating uncertainty estimation (Giering et al., 2019; Hans et al., 2019), which yield similar results (<0.01 mm difference). The SST algorithm involves a localized approach, requiring Monte Carlo simulation. Algorithmic uncertainties were further estimated by comparing retrievals with simulated truth from the microwave radiative transfer model, yielding standard uncertainties of 0.51 mm for CWV and 0.20 K for SST. These values are lower than those reported by Wentz (2000), due to enhanced channel selection and algorithm design. Final level-2 uncertainties (Table 5) were obtained by combining both components using propagation of uncertainty (Giering et al., 2019; Hans et al., 2019).

For level-3 gridded products, retrieval uncertainties are substantially reduced through spatial and temporal averaging. In this study, we focus primarily on the statistical uncertainty (SU) associated with trend estimation, which is especially relevant given the limited record length and the presence of low-frequency climate variability (Zou et al., 2023). Following the approach of Santer et al. (2008), we account for temporal autocorrelation in the anomaly time series by adjusting the effective sample size using the lagged autocorrelation coefficient. This correction ensures that the derived confidence intervals for linear trends are not underestimated, providing a more reliable assessment of long-term climate signals. The SU estimates of the trends are expressed at the 95% confidence level, is computed as:

$$SU = 1.96 \cdot SE \cdot \sqrt{\frac{1+r}{1-r}}, \qquad (9)$$

where $SE$ is the uncorrected standard error from ordinary least squares regression, and $r$ is the lagged autocorrelation coefficient. Unless otherwise specified, $r$ refers to the correlation between values that are 1 time period apart (lag-1 autocorrelation) throughout this study. The level-3 trend uncertainty is not included in Table 5, but is reported in all trend analysis in the relevant comparison figures below.

## 4 Validation Results

### 4.1 Validation with Radiosonde and Buoy observations

Figure 5 shows scatter diagrams of the CWV and SST retrievals against in-situ measurements for each of the AMSR-E, AMSR2, and MWRI instruments. Collocated samples span the entire lifetime for AMSR-E and AMSR2, and from 2011 to 2015 for MWRI, when its LECT drifted less than 1 h (Wu et al., 2020).

For CWV, the satellite retrievals from all three instruments are in good agreement with the RAOB sounding observations (Fig. 5a–5c) and exhibit similar statistical characteristics. Specifically, the retrieval bias is generally less than 0.1 mm, and the root

mean square errors (RMSEs) are about 4 mm for all three instruments. This accuracy is comparable to other retrieval products from satellite-based instruments such as the Special Sensor Microwave Imager (SSM/I) and the AMSR-E (Deeter, 2007; Liu et al., 2023). This indicates that there is no significant difference between the retrievals from the three instruments using the same algorithm. It is worth noting that some of the residual variability may originate from collocation and representation mismatches (Zou et al., 2006; Giering et al., 2019; Hans et al., 2019). Radiosonde ascents typically take up to 2 h and can drift

over 100 km (Ingleby et al., 2016). Moreover, satellite-derived CWV includes the full atmospheric column, whereas RAOB may miss water vapor near the surface depending on station elevation (Buehler et al., 2012).

For the SST CDR, retrievals from all three instruments show similar accuracy compared to SST$_{GODAE}$ (Fig. 5d–5e), with biases and RMSEs within 0.2 K and 1.6 K, respectively. This result is also similar to other retrievals from AMSR-E, AMSR2, and MWRI (Gentemann, 2014; Gentemann and Hilburn, 2015; Liao et al., 2017; Li and Jiang, 2024). The observed differences

can be partially attributed to representation differences between microwave and in situ measurements. While satellite sensors retrieve sub-skin temperature at millimeter depth, GODAE integrates in situ measurements from ships and buoys, primarily at a depth of 1 meter (Huang et al., 2020).

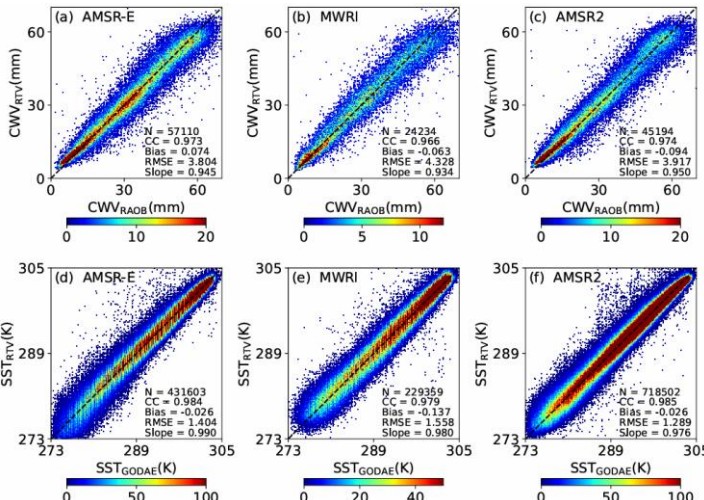

**Figure 5: Density scatter diagram of retrieved CWV (top) and SST (bottom) versus in-situ measurements (CWV$_{RAOB}$ and SST$_{GODAE}$).**
**From left to right: AMSR-E, MWRI, and AMSR2. The black dotted lines represent the diagonal. (N: Sample number; CC: Correlation coefficient; RMSE: Root mean square error; Slope: Slope of linear regression).**

## 4.2 Validation of long-term climate trends against ERA5, RSS retrievals, GTMBA, and Ground GNSS observations

As one of the most important applications of CDRs, the capability of retrieved CWV and SST to detect climate trends is assessed here. The first evaluation is a comparison with long-term ground-based GNSS observations. As a ground-based observation system, GNSS is completely independent of retrievals and offers a reliable local reference. Such a comparison provides insight into the CDR trends at local geolocations. Figure 6 shows time series of monthly mean CWV anomalies from our merged satellite retrievals ($CWV_{RTV}$) and GNSS observations at three selected sites: BRMU, COCO, and DGAR. For all sites, the time variation of the GNSS CWV anomalies is highly consistent with $CWV_{RTV}$, with a statistically significant correlation coefficient (CC) of more than 0.91. In addition, comparisons with RSS and ERA5 at these three sites are also shown in Fig. 6. Similar correlations were found between $CWV_{RTV}$ and the ERA5 and RSS datasets. The $CWV_{RTV}$ trends at the three sites vary from 0.4 to 1.2 mm decade$^{-1}$, suggesting a large range and complexity in trend variations at different spatial locations. For all three sites, trend differences between $CWV_{RTV}$ and GNSS observations range from 7% to 20%. This is better than the ERA5 and RSS validations, where trend differences can reach 50%–80% when compared to GNSS observations at some sites.

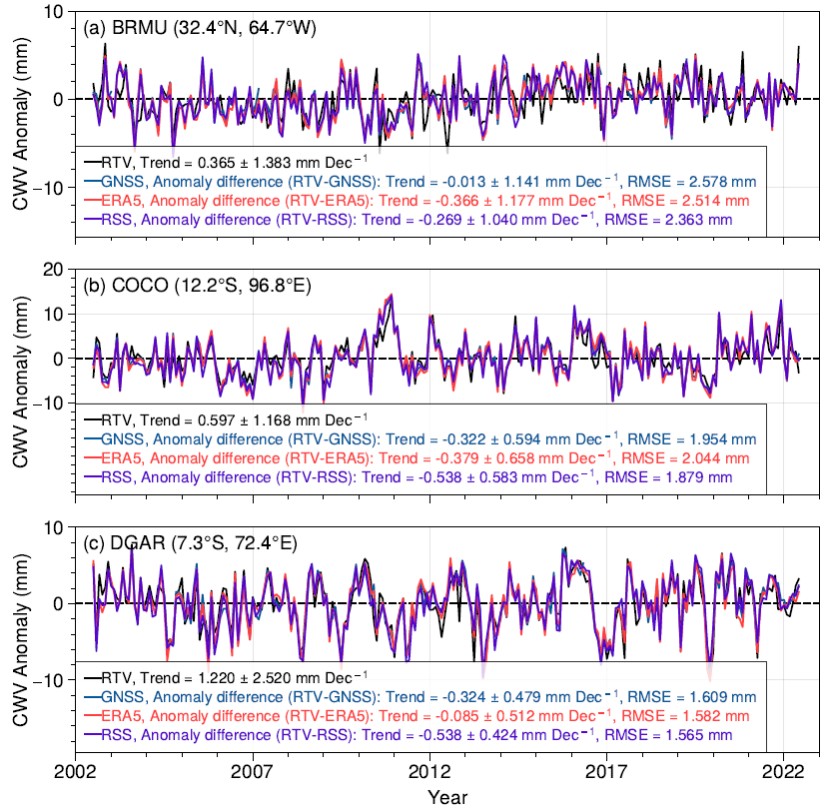

**Figure 6: Monthly anomaly time series for CWV$_{RTV}$, GNSS, ERA5, and RSS over the three GNSS sites. The legend in each panel includes statistical metrics for trend comparison. The black text indicates the linear trend of the RTV anomaly (black line) along with its SU. Colored text represents the trend values, SUs, and RMSEs of the anomaly differences between RTV and each of the validation datasets (GNSS, ERA5, and RSS). As the anomaly differences are computed over identical time periods, the mean bias is inherently zero and thus not shown. The lag-3 autocorrelation is used for the BRMU station in a).**

Figure 7 presents the time series of monthly mean anomalies for $SST_{RTV}$ and other products for the three GTMBA sites. Since GTMBA is a moored in situ buoy network, it is fully independent of the satellite retrievals and serves as a reliable reference for evaluating SST trends at fixed ocean locations. Similar to the CWV, their time series exhibit comparable seasonal variations, with correlations above 0.75, statistically significant at the 99% significance level. The trends of $SST_{RTV}$ are within 0.3 K decade[-1], and its trend differences with GTMBA are within 0.11 K decade[-1].

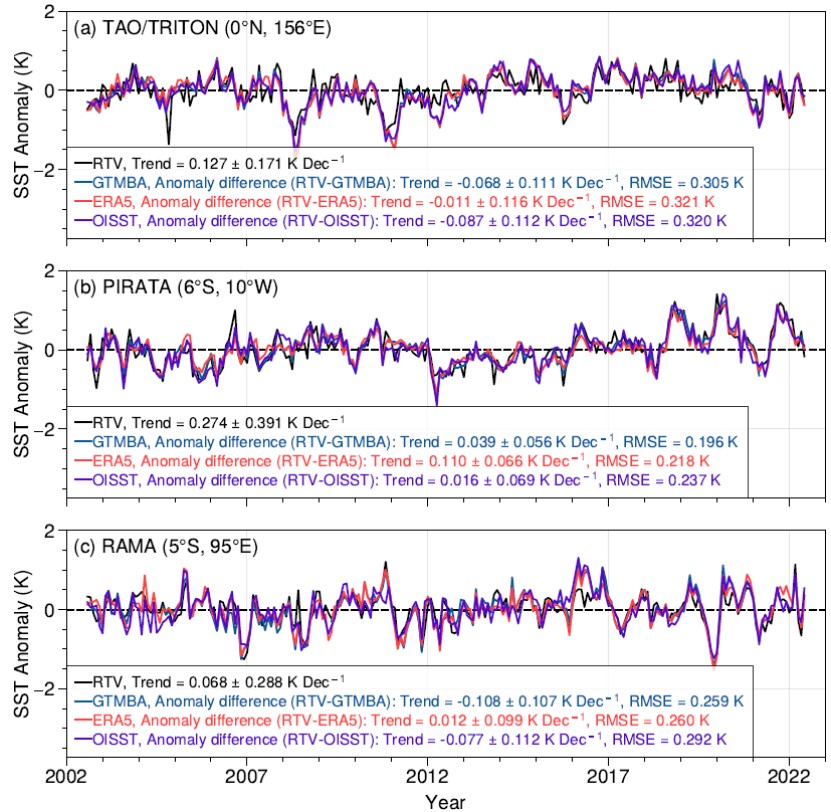


**Figure 7: Similar to Fig. 6, but for $SST_{RTV}$, GTMBA, OISST and ERA5 data at three GTMBA sites.**

Figure 8a compares trends for global ocean monthly anomalies between our CWV retrievals and ERA5 and RSS, while Fig. 8b presents SST comparisons between our retrievals and ERA5 and OISST. For CWV, all products are closely correlated with the AMSR-E and AMSR2 measurements. Specifically, the observed AMSR-E and AMSR2 TBs were assimilated into ERA5

with bias corrections (Dee, 2005; Kazumori et al., 2016), while our $CWV_{RTV}$ and the RSS datasets are retrieved and merged products from AMSR-E and AMSR2 TB measurements (Mears et al., 2018). This is demonstrated by their high CCs exceeding 0.94. For long-term trends, $CWV_{RTV}$ exhibited a positive trend of 0.39 mm decade[-1], which is close to the ERA5 and RSS values of 0.41 and 0.40 mm decade[-1], respectively. For SST, all products exhibited a large-scale warming trend, with values of 0.16, 0.15, and 0.20 K decade[-1] for the $SST_{RTV}$, ERA5, and OISST datasets, all with CCs exceeding 0.91.

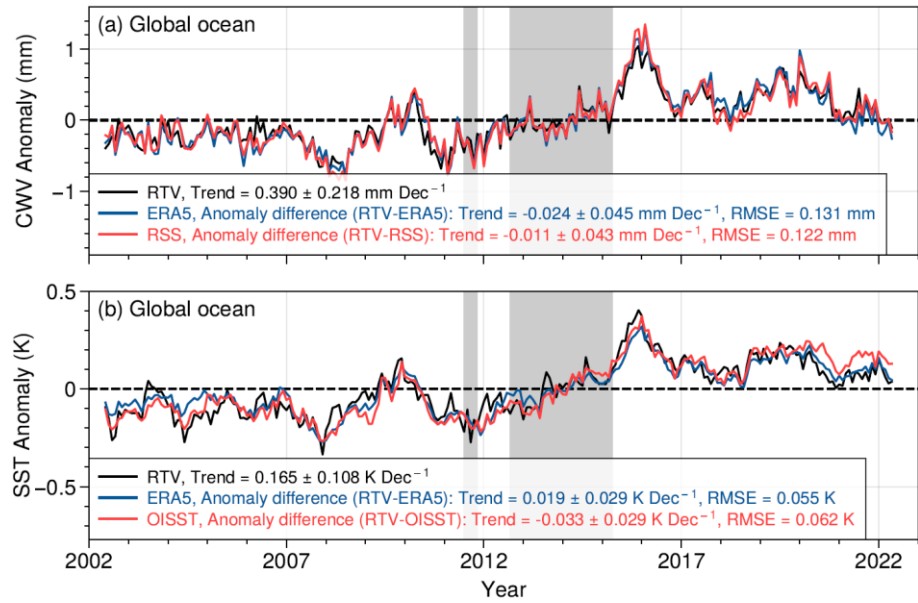


**Figure 8: Monthly anomaly time series of (a) CWV and (b) SST for different datasets over global oceanic areas from 2002 to 2022. The legends indicate the linear trends and their SUs for the RTV anomalies (black lines) as well as for the anomaly differences between RTV and each validation dataset. The grey areas represent the overlap periods of MWRI with AMSR-E or AMSR2.**

Although both RSS and ERA5 include AMSR-E and AMSR2 data, their processing chains are entirely independent of the

RTV. As a peer satellite climate product, RSS uses a different retrieval algorithm and merges the measurements from additional sensors (e.g., SSM/I and TMI) but not MWRI (Mears et al., 2018). ERA5 also excludes MWRI and applies variational bias correction to remove satellite drifting biases, and assimilates observations into a model-based reanalysis system (Hersbach et al., 2020). These differences make them suitable, independent references for evaluating variability and trends. Although independent, including the fully independent OISST, all datasets exhibited a shift from 2010 to 2016 (Fig. 8). This most likely

suggests that this shift is a true climate shift.

We further compare trends between different datasets of CWV and SST from 2003 to 2020 over the tropical ocean (20°S–20°N) (Fig. 9), where there is well-understood covariability between temperature and atmospheric moisture (Wentz and Schabel, 2000; Held and Soden, 2006; Mears et al., 2007). The RSS-CDR trends is also included during this period. Figure 9a shows that all products exhibit similar interannual variations, with more pronounced amplitudes than those of the global ocean.

Trend values for CWV during 2002–2022 are 0.61, 0.81, 0.81, and 0.83 mm decade$^{-1}$ (1.49%, 1.97%, 1.97%, 2.02% decade$^{-1}$) for the $CWV_{RTV}$, ERA5, RSS, and RSS-CDR datasets, respectively. Meanwhile, the SST trends for the same period are 0.15, 0.14, 0.23, and 0.14 K decade$^{-1}$ for the $SST_{RTV}$, ERA5, OISST, and RSS-CDR datasets, respectively. The differences of these trend values are essential in analyzing the covariability between CWV and SST over the tropical ocean, which will be discussed in Sect. 5.

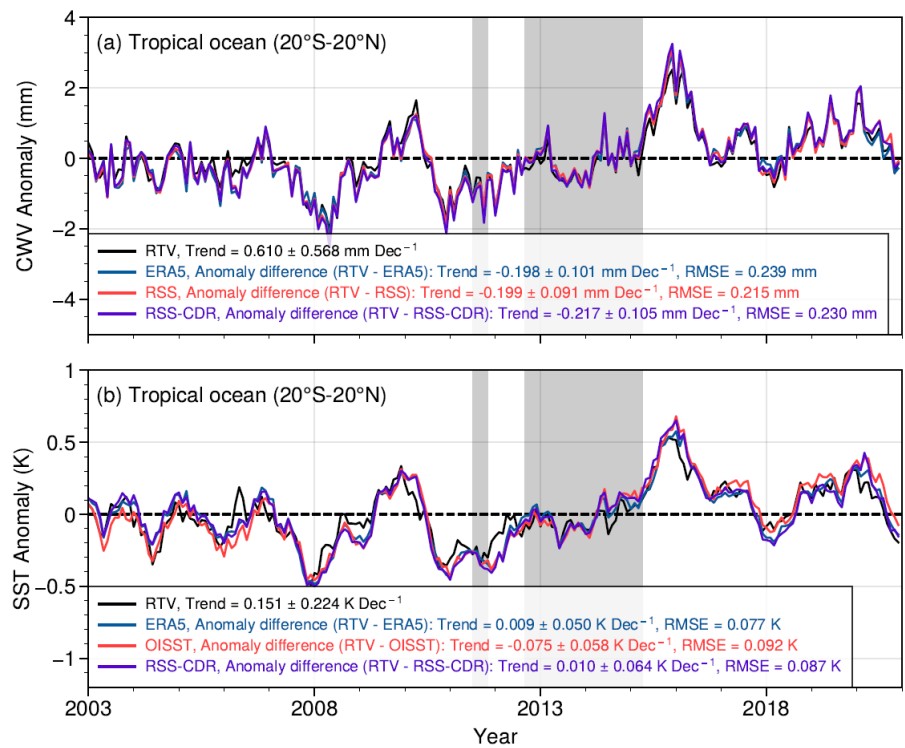


**Figure 9: Similar to Fig. 8, but for the tropical ocean (20˚S–20˚N) from 2003 to 2020.**

The global trend distribution for CWV$_{RTV}$ and SST$_{RTV}$, along with those from ERA5, RSS, and OISST, is shown in Fig. 10. Different datasets show overall similar trend patterns for both CWV and SST, with spatial CCs exceeding 0.78 and 0.85, respectively. As a result of human-induced global warming, most global oceanic regions (80.7% of grids) show a positive

trend in CWV, including the tropical Indian Ocean, the eastern North Pacific Ocean, the South Pacific Ocean, and the western tropical Pacific Ocean, with all of these regions passing the 99% significance test. This trend pattern is consistent with other studies (Wang J. H. et al., 2016; Wang Y. et al., 2016; Adler et al., 2017). The regions with negative trends are mainly located along the equatorial central Pacific to the west coast of South America. The SST warming regions are consistent with the positive CWV trend regions and also pass the significance test. This is consistent with their physical covariance, as determined

by the Clausius-Clapeyron equations (Santer et al., 2021; Held and Soden, 2006).

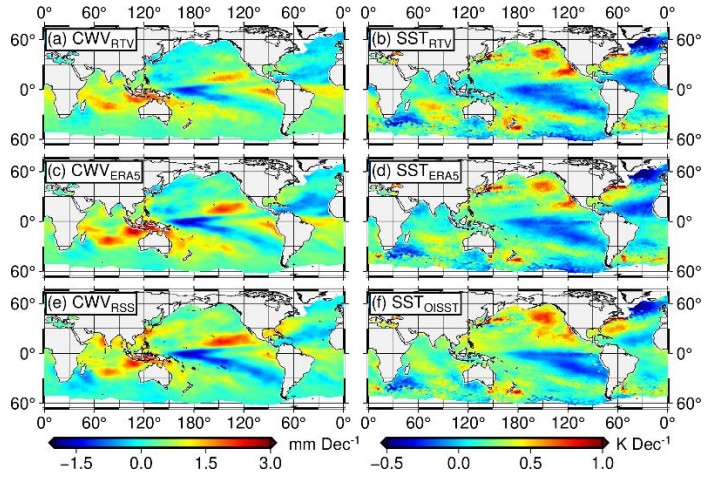

**Figure 10: Global trend distribution for CWV (left) and SST (right) from 2002 to 2022. From top to bottom: CDR_RTV, ERA5, and RSS or OISST.**

## 5. Trend Covariance in CWV and satellite temperature products over the Tropical Ocean

As mentioned earlier, the Clausius-Clapeyron relationship imposes a strong thermodynamic constraint on the covariance of CWV and temperature products under long-term warming. This physical linkage can be used to examine whether the joint behaviour of independently retrieved CWV and temperature variables conforms to expectations under radiative climate forcing. In fact, Santer et al. (2005, 2021) pointed out that this approach provides a meaningful way to evaluate observational datasets, with climate models offering physically grounded reference values. Following this rationale, we regard the trend ratios between

CWV and SST (or TLT, TMT) as physically interpretable metrics that support mutual consistency checks between observations and model simulations.

Figure 11a compares the ratio of CWV to SST ($R_{\{CWV/SST\}}$) over the tropical ocean (20°S–20°N) from our retrieved products with those from the CMIP6 multi-model simulations, as well as ERA5 and RSS-CDR products. To ensure a fair evaluation, both CWV and SST products were sourced from the same groups — either RSS-CDR, ERA5, or our retrievals. Although

individual trends in CWV or SST differ significantly among different CMIP6 models, their ratios, $R_{\{CWV/SST\}}$, consistently align around the 8.6% $K^{-1}$ regression diagonal line (Fig. 11a). This occurs because most CMIP6 model simulations follow the strict Clausius-Clapeyron physical constraints in an adiabatic moisture process (Santer et al., 2021), providing a strong constraint on the observed ratio. For the retrieved CWV and SST products in this study, the $R_{\{CWV/SST\}}$ ratio is 9.9% $K^{-1}$, quite close to the CMIP6 regression value of 8.6% $K^{-1}$. On the other hand, the $R_{\{CWV/SST\}}$ ratios for the ERA5 and RSS-CDR

products are as high as 13.9% and 14.3% $K^{-1}$, respectively. A similar high $R_{\{CWV/SST\}}$ was found over a longer time period for the RSS-CDR dataset (Santer et al., 2021). The much larger ratio in the ERA5 and RSS-CDR datasets occurred because their

CWV trends appeared to be too large, while SST trends were slightly smaller compared to those in our retrieved CDR over the tropical ocean (Fig. 9, Fig. 11a).

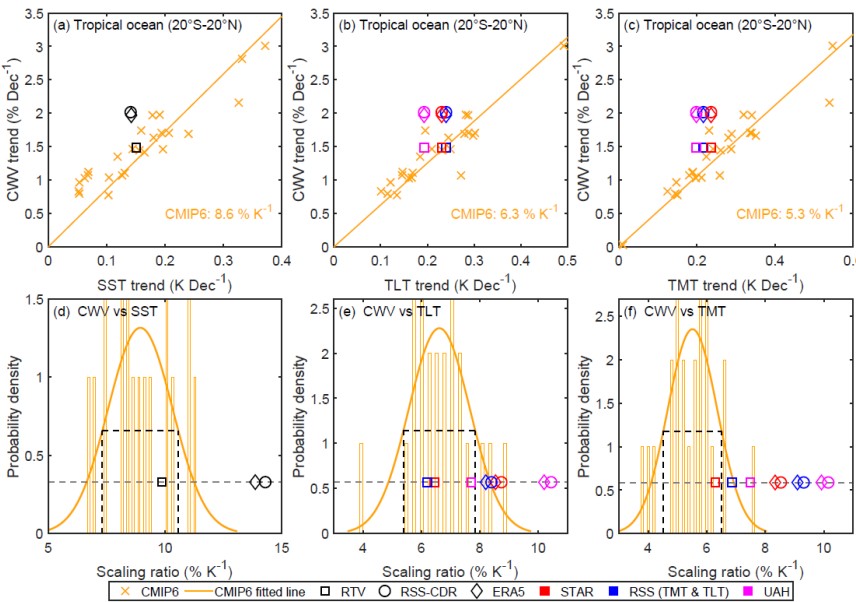

**Figure 11: Comparison of trend covariance over the tropical ocean (20° N–20° S) during 2003–2022 showing scatterplots of tropical trends in (a)–(c) and histograms of the trend ratios between observations and CMIP6 model simulations in (d)–(f). The three comparisons from left to right are CWV versus SST, CWV versus TLT, and CWV versus TMT. In (a)–(c), the orange "x" symbols represent trend ratios from CMIP6, and the straight lines are their fitted lines. Squares, circles, and diamonds represent the CWV and SST from RTV, RSS-CDR, and ERA5, respectively. In (b), (c), (e), and (f), the different colours of symbols indicate the CWV**
**trend plotted against observed TLT or TMT trends from STAR (red), RSS (blue), and UAH (purple). In (d)–(f), the orange curves are the fitting curves of the Gaussian distribution for CMIP6, and the black dashed line represents the fitted FWHM. Observational trend ratios are plotted in one reference row (grey dashed line), whose y-axis offset does not represent the actual value.**

To further assess the accuracy of the $CWV_{RTV}$ trend, we pair it with trends from atmospheric layer temperatures observed by satellite microwave sounders (Fig. 11b and 11c), following the studies in Santer et al. (2021). For ratios of CWV over TLT
($R_{\{CWV/TLT\}}$) and TMT ($R_{\{CWV/TMT\}}$), the combination of $CWV_{RTV}$ with the STAR dataset is closest to the fitted straight line (Fig. 11b and 11c), with values of 6.3% $K^{-1}$ and 6.5% $K^{-1}$, compared to the CMIP6 values of 6.3% $K^{-1}$ and 5.3% $K^{-1}$, respectively. The combination of $CWV_{RTV}$ and RSS-CDR TLT also gives a ratio of $R_{\{CWV/TLT\}}$ that is closer to the expected CMIP6 value. The STAR TMT shows a high accuracy of 0.012 K decade$^{-1}$ in trend detection from 2002 to 2022, as it was developed based on satellite microwave sounder observations in stable orbits with high radiometric stability (Zou et al., 2018,
2021). The fact that the combination of $CWV_{RTV}$ with STAR TMT is closest to the expected CMIP6 value strongly suggests that the our $CWV_{RTV}$ product also has high accuracy in trend detection. In contrast, the CWV trends in ERA5 and RSS-CDR appeared to be too high, causing the $R_{\{CWV/TLT\}}$ and $R_{\{CWV/TMT\}}$ ratios to deviate significantly from the expected ratios in the CMIP6 simulations.

Figures 11d–11f show the histogram distributions of the CMIP6 simulated ratios from different models for $R_{\{CWV/SST\}}$,
$R_{\{CWV/TLT\}}$, and $R_{\{CWV/TMT\}}$, compared with ratios derived from other data products. The full width at half maximum (FWHM)

of the Gaussian fitting distribution defines a range within which ratios from observational products are consistent with the CMIP6 simulations. Their consistencies are further evaluated using the Z-score (Trenberth et al., 2005), which is calculated as the difference between the trend ratios of observed data products (RTV, ERA5, or RSS-CDR) and the CMIP6 model simulations, divided by the standard deviation of all CMIP6 models. A lower Z-score indicates better agreement with climate model simulations. In Fig. 11d, the retrieved CWV and SST products in this study fall within the FWHM, with Z-scores of 0.67. This performs much better than the other products, which have Z-scores above 3. In Fig. 11e, the combination of $CWV_{RTV}$ with TLT from the three groups—STAR, RSS, and UAH—are all within the FWHM range, with the former two products positioned in the middle of the FWHM. For the ratio of $R_{\{CWV/TMT\}}$ (Fig. 11f), only the combination of $CWV_{RTV}$ and STAR TMT falls within the FWHM, with a Z-score of 0.92. These results suggest that ratios from our retrieved products, combined with satellite observations in stable orbits, generally align with expectations from the CMIP6 model simulations.

The overall agreement in the $R_{\{CWV/SST\}}$ ratios between climate model simulations and our retrieved products, as well as the combined satellite microwave imagery and sounding products, provided an extremely encouraging result. This highlights the value of using climate covariance to evaluate the internal consistency of both observations and models. When retrieved satellite products reproduce the trend ratios expected from CMIP6 simulations, it increases confidence in their physical reliability. It also suggests that past disagreements between models and observations may have stemmed from biases in earlier datasets, rather than flaws in the models themselves. With careful calibration and inter-calibration, satellite observations not only follow the constraints from climate model simulations, even when observations are from completely different instrument sources, but also provide a constraint on climate model simulations regarding trends in separate CWV and SST.

## 6. Data availability

The CDR for CWV and SST described in this work is available from the Zenodo repository: https://doi.org/10.5281/zenodo.14539414 (Fu et al., 2024).

## 7. Conclusion

We developed a set of CDRs for CWV and SST spanning over 20 years, from 2002 to 2022. The dataset is based on recalibrated AMSR-E, MWRI, and AMSR2 radiance measurements. A diurnal drift adjustment was conducted on the MWRI radiances using novel diurnal adjustment algorithms developed by Zou et al. (2023). The adjustment ensures consistency in the developed CDRs and greatly enhances their application value in climate research. The retrieval algorithm utilizes the logarithmic relationship between observed TBs from multiple satellite microwave imagery channels and CWV and SST. The accuracy and long-term trends of the CWV and SST CDRs were validated against various observations, including in-situ data, satellite retrievals, and climate reanalyses.

In general, the retrieved CWV and SST are in good agreement with in-situ observations, with small biases and RMSE, and are consistent with retrievals from other instruments. The variability in the developed CDR time series shows high correlation when compared with other observations, including GNSS, GTMBA, RSS, ERA5, and OISST. Long-term trends of the presented CDRs are generally consistent with the ERA5, RSS, and OISST datasets on a global scale. The most encouraging result is that the covariance between our retrieved CWV and SST over the tropical oceans is close to the expectations from

CMIP6 model simulations. The ratios of our retrieved CWV to the layer mean temperatures from satellite microwave sounder observations also show favorable agreement with expectations from climate model simulations. Given the constraint on the CWV and SST trend ratios provided by CMIP6 model simulations, these agreements suggest good accuracy in trend detection by the retrieved CWV and SST products. In turn, because the satellite CWV and SST are from well-calibrated satellite observations of different instrument sources, the agreement also provides a constraint on trends in separate CWV and SST

from climate model simulations.

All these evaluation results suggest that the CDRs developed in this study can be effectively applied to climate change research due to their high consistency, accuracy, and continuity.

In the future, as more satellites are launched and microwave radiometers continue to be introduced and recalibrated for the generation of FCDR datasets, it will be possible to establish CDRs for various geophysical parameters with longer time spans

and higher sampling frequencies. In particular, we plan to extend the CWV record back to 1987 using SSM/I and the SST record back to 1997 using TMI, respectively. Meanwhile, the forward extension will include ongoing AMSR2 observations and data from next-generation MWRI instruments onboard subsequent FY-3 series satellites. This will enable coverage of the standard 30-year reference periods defined by the World Meteorological Organization (WMO), ensuring that the dataset remains comprehensive. This will help to better characterize climate change and the diurnal variation of environmental

variables, and improve our understanding of the mechanisms behind them.

**Author contributions**

Conceptualization, YW; methodology, YF, CZZ, PZ, BW, SL, YW; investigation and analysis: YF, CZZ, YW, SW; funding acquisition, YW; writing (original draft preparation): YF, CZZ, PZ, YW. All authors have contributed to the draft.

**Competing interests**

The contact author has declared that none of the authors has any competing interests.

**Acknowledgements**

The authors would like to thank NSIDC, China Meteorological Administration (CMA), Japan Aerospace Exploration Agency (JAXA), NOAA, RSS, FNMOC, ECMWF, and World Climate Research Programme (WCRP) for providing standard products for this study. This work was supported by the National Natural Science Foundation of China (grant 41875024 and grant
42075124); Anhui Provincial Key Research and Development Project (grant 2408055UQ005); The China Meteorological Administration Science the Technology Projects (grant CMAJBGS202502); Ministry of Science and Technology of China (grant 2022YFC3104303); the FY Satellite Advanced Planning (grant FY-APP-ZX-2022.0212); Civil aerospace "14th Five-Year" pre-research project of State Administration of Science, Technology and Industry for National Defense.

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
