# Peer review of "A climate data record of atmospheric moisture and sea surface temperature from satellite observations"

_Earth System Science Data, 2024_

## Referee Comment (RC2)

**Review of https://doi.org/10.5194/essd-2024-608 A climate data record of atmospheric moisture and sea surface temperature from satellite observations**

By: Yixiao Fu, Cheng-Zhi Zou, Peng Zhang, Banghai Wu, Shengli Wu, Shi Liu, and Yu Wang

**Overall Recommendation**

This paper presents a new climate data record of column water vapor and sea surface temperature from microwave instruments covering the period 2002-2022. The quality of the CDR was evaluated against several reference datasets, i.e., radiosonde, radio occultation, and alternative remote sensing-based retrievals, reanalysis, and climate model simulations. The evaluation results confirm that the data record meets the specifications of a climate data record in terms of bias and noise against reference observations and trends against reanalysis and model simulations.

In general, the manuscript is well understandable and complete. The manuscript is very exhaustive in terms of references to other paper and use of evaluation data. Still, the paper can be written more focused and to the point. Right now, it not fully clear how all information is linked together. To address this, the authors could, among others:

- Provide a pointwise description of what validation metric is evaluated with what reference data.
- Explain to what extent the reference data are independent of, and superior to, the CDR.
- Explain to what extent the CWV and SST products are independent of each other.
- Explain how uncertainties from input data and processing steps propagate into the CDR product
- Discuss, and if possible assess, the role of validation uncertainties arising from collocation, synchronization, and representation differences between the reference data and the CDR.

The manuscript needs minor revisions before it can be published. The following are some general criticisms followed by a chronological list of minor criticisms

**General criticisms**

**Independence of the WV and SST products**

To what extent can the WV and SST data be considered independent of each other. Both retrievals are based on TB data from the same instruments and same FCDR, are these products not consistent by design? I expect the difference is in the fact that CWV is based on cloudy cases and SST based on cloud free cases. Thus, an important aspect in the retrievals is the cloud decision, which is coming from the cloud parameter input. Please comment and explain in text more on the role of the cloud parameter input in the retrieval.

**Independence of the validation data**

To what extent are the validation data independent of the CDR data, please explain.

**Validation Data**

Different source of in-situ validation data is used. The authors should make clearer that these are of superior quality, independent, and consistent with each other.

Especially, I am sceptical about using CMIP simulations as validation data. To underpin the value of using CMIP simulations in this article, the authors should clearly explain for what statistical metrics the CMIP simulations contribute to the validation of observations.

The authors correctly indicate that the CWV and SST variables of CMIP relate to each other by design, via the Clausius Clapeyron law. This predetermined relationship makes that one cannot use both variables as independent sources of validation data. In the paper only one variable can be used for validation. Please comment on this.

**Uncertainty and error propagation**

The authors spend little words on the uncertainty of the CWV and SST products, and do not provide uncertainty bars in their figures. Literature describes approaches to estimate uncertainties of level 1 and level 2 satellite data, using metrological principles. Relevant work on this was done in the framework of the FIDUCEO project

**FIDUCEO method paper**

Giering, R.; Quast, R.; Mittaz, J.P.D.; Hunt, S.E.; Harris, P.M.; Woolliams, E.R.; Merchant, C.J. A Novel Framework to Harmonise Satellite Data Series for Climate Applications. *Remote Sens.* **2019**, *11*, 1002. https://doi.org/10.3390/rs11091002

**FIDUCEO example paper**

Hans, I.; Burgdorf, M.; Buehler, S.A.; Prange, M.; Lang, T.; John, V.O. An Uncertainty Quantified Fundamental Climate Data Record for Microwave Humidity Sounders. *Remote Sens.* **2019**, *11*, 548. https://doi.org/10.3390/rs11050548

More general descriptions for performing error propagation for Essential Climate Variables (ECVs) of the Global Climate Observing System, involving systematically tracking and quantifying uncertainties through all stages of the data processing, i.e., from raw observations to final climate data products, are, described in the following papers.

Roebeling, R. A., S. Bojinski, P. Poli, V. O. John, and J. Schulz, 2025: On the Determination of GCOS ECV Product Requirements for Climate Applications. *Bull. Amer. Meteor. Soc.*, **106**, E868–E893, https://doi.org/10.1175/BAMS-D-24-0123.1.

Merchant, C. J., Paul, F., Popp, T., Ablain, M., Bontemps, S., Defourny, P., Hollmann, R., Lavergne, T., Laeng, A., de Leeuw, G., Mittaz, J., Poulsen, C., Povey, A. C., Reuter, M., Sathyendranath, S., Sandven, S., Sofieva, V. F., and Wagner, W., 2017: Uncertainty information in climate data records from Earth observation, Earth Syst. Sci. Data, 9, 511–527, https://doi.org/10.5194/essd-9-511-2017

**Climate data records temporal coverage**

Climate studies often ask for data that covers the standard 30-year reference period defined by WMO, eg 1991-2020 (current reference period) or 2000-2030 (next reference period). Are there any plans to expand further back in time, e.g., back to 1997 by using TMI data, to cover the next reference period?

**Minor criticisms**

*Section 3: CDR Data:* Add a table listing what data were used to construct the CDR, ie, name instrument the start date, end date, and discuss what steps were taken to harmonize and homogenise (see definitions https://research.reading.ac.uk/fiduceo/glossary/) these data over the entire CDR period, so as to use them for trend analysis

| Satellite | Instrument | Start date | End date |
|-----------|------------|------------|----------|
| FY3b      | MWRI       | 2010       | 2021     |
| Aqua      | AMSR-E     | 2002       | 2016     |
| GCOM-W1   | AMSR2      | 2012       | 2025     |

Page 9, line 232: replace "thereby eliminated" by "thereby empirically correcting"

Equation (1), (4), (5): For consistency with the other notations, can you replace

D(X,m,L), TB(X,t,m,L), and TB'(X,t,m,L)with  $D_{MWRI}(X,m,L)$  and  $TB_{MWRI}(X,t,m,L)$ , and  $TB'_{MWRI}(X,t,m,L)$

*Figure 3:* is the blue line in this figure not the difference of the two instruments before correction, thus:

 $TB_{MWRI}(X,t,m,L) - TB_{AMSR}(X,t,m)$ Instead of  $\Delta TB' = TB'_{MWRI}(X,t,m) - TB_{AMSR}(X,t,m)$

This is what I expect, because then the figure demonstrated that the difference between the instruments before corrections resemble those of the diurnal anomalies and thus seems to prove that a correction is needed. Please explain.

*Figure 6 (and similar figures later):* Indicate in the caption that the numbers given in the legend for each validation result represent BIAS and RMSE

*Figure 8 & 9:* The compared datasets are partly based on the same observational datasets (AMSR-E and AMSR2) and thus cannot be considered independent of each other. I realize that complete independence is difficult to achieve, still the authors discuss and provide evidence of the degree of independence of the compared datasets. This is

especially important in determining the climatological significance of jumps in the time series.

*Figure 8 & 9:* With reference to my above point, there is a clear jump in values between the period 2002-2012 and 2016-2022. This jump seems rather to be related to a change in instrument than to a change in climate. Please comment.

*Line 375:* The statement "*Different datasets show overall similar trend patterns for both CWV and SST.*" is very qualitative. Can you provide some statistics to make it more quantitative.

*Figure 11:* Why are there more crosses for the CMIP trends. Are these different CMIP scenarios?

*Line 452 Conclusions:* Is the statement "*The most encouraging result is that the covariance between our retrieved CWV and SST over the tropical oceans is close to the expectations from CMIP6 model simulations.*" true?

CMIP is an ensemble of model simulations, matching with CMIP does not say much about the quality of the observational data. May be one could reason the other way around and write that it is encouraging that the CMIP simulations seem to be able to reproduce the observed relationships. This would, however, be a statement about the quality of CMIP and not about the quality of the CWV and SST observations! Please comment.

---

## Author Comment (AC1)

**Reply to Comments from Anonymous Referee #1**

1. Line 201: "The synthetic radiance dataset, consisting of these recalibrated observations from different sensors, is referred to as the FCDR (Liu et al., 2023; Poli et al., 2023)." - these are not synthetic data, but actual measurements. Synthetic data for example in Poli et al., 2023 refers to simulated measurements using a radiating transfer model from reanalysis outputs for instance.

**Reply: We thank the Anonymous Referee for this important correction. We agree that the use of the term "synthetic" was inaccurate in this context. Accordingly, we have revised the sentence in Line 201 as follows.**

*Section 3, Lines 212-213*: "The continuous and consistent radiance dataset, consisting of recalibrated observations from different sensors, is referred to as the FCDR (Liu et al., 2023; Poli et al., 2023)."

**Reply to Comments from Anonymous Referee #2**

**Reply: We sincerely thank the anonymous referee for the thorough and constructive review. We have carefully addressed all comments, as outlined below. In the following, we provide responses in the order of the General Criticisms, Minor Criticisms, and Overall Recommendation sections, respectively, as presented in the anonymous referee's report.**

**General Criticisms**

1. *Independence of the WV and SST products*. To what extent can the WV and SST data be considered independent of each other. Both retrievals are based on TB data from the same instruments and same FCDR, are these products not consistent by design? I expect the difference is in the fact that CWV is based on cloudy cases and SST based on cloud free cases. Thus, an important aspect in the retrievals is the cloud decision, which is coming from the cloud parameter input. Please comment and explain in text more on the role of the cloud parameter input in the retrieval.

**Reply: We appreciate the Anonymous Referee's insightful comment. Contrary to the assumption that SST is retrieved only under clear-sky conditions, both CWV and SST in our dataset are retrieved under rain-free but not necessarily cloud-free scenes. The retrieval algorithms, following Wang et al. (2009), are robust under moderate cloud contamination. As both products are derived from the same FCDR of TBs, they share consistent temporal and spatial sampling, which facilitates joint variability analysis.**

**Despite the common data source, CWV and SST retrievals are independent by design. Fundamentally, the radiometer channels are engineered such that measurements at different frequencies can be used to obtain information about different ECVs. For example, those low-frequency channels like 10V and 10H are primarily sensitive to SST, while higher-frequency channels such as 18H, 23V, and 36H are more responsive to CWV. In the retrieval process, CWV is estimated using a first-order regression, whereas SST is retrieved using a second-order regression with regionally optimized coefficients. These differences in spectral sensitivity and algorithm formulation ensure that CWV and SST are physically and mathematically independent, even though they are retrieved from the same instrument observations and FCDR. This distinction is now clarified in the revised manuscript. In addition, we have added the initial estimated coefficients associated with used channels and their sensitivities for SST retrieval in Table 4 to demonstrate the differences in CWV and SST retrievals.**

*Section 3, Lines 315–319*: "From the perspective of instrument design and microwave radiative transfer, different ECVs respond to measurements of different frequency channels in distinct ways. As shown in Table 4, low-frequency channels such as 10V and 10H are primarily sensitive to SST, while higher-frequency channels like 19H, 23V, and 37H are more responsive to CWV. As a result, although CWV and SST are both retrieved from the same set of satellite TBs, the differences in channel sensitivity and retrieval algorithm expressions result in them being physically and mathematically independent products."

**Table 4. Regression coefficients and the corresponding sensitivities of CWV and SST to TBs for different PMW channels used in the retrieval algorithms. For SST, the regression coefficients represent the initial estimate.**

|  |  | Offset | TB 10V | TB 10H | TB 19H | TB 23V | TB 37H |
|---|---|---|---|---|---|---|---|
| **CWV** | $a_i$ | -9.953 | 42.890 | 29.689 | -84.776 | -44.691 | 41.433 |
| **(mm)** | $\Delta$ CWV/$\Delta$ TB |  | -0.387 | -0.151 | 0.524 | 0.732 | 0.319 |
| **SST** | $b_i$ | -559.777 | -940.886 | -406.252 | 201.324 | -72.084 | 47.956 |
| **(K)** | $c_i$ |  | -393.786 | -637.333 | 188.989 | -20.560 | 13.612 |
|  | $\Delta$ SST/$\Delta$ TB |  | 1.764 | -0.313 | 0.082 | 0.118 | -0.203 |

2. *Independence of the validation data.* To what extent are the validation data independent of the CDR data, please explain.

**Reply: We thank the Anonymous Referee for raising this important point. Among all CWV and SST products, RAOB, GTMBA buoys, GNSS station CWV, and OISST SST are fully independent of our CDR, as they originate from distinct measurement systems and are not involved in any stage of our retrieval or calibration. These in situ datasets serve as objective references for assessing retrieval accuracy.**

The RSS CWV product, although also based on AMSR-E and AMSR2, is a peer satellite climate product developed with different recalibration approaches and retrieval algorithm, incorporating additional observations from other sensors such as SSM/I, TMI, and GMI. Importantly, MWRI is not included as a bridge instrument that can fill the gap in AMSR-E and AMSR2 observations, which is a key component of our CDR. Therefore, RSS product serves as an independent algorithmic reference data for evaluation. In terms of the ERA5 reanalysis, AMSR data is used for assimilation with variational bias correction to account for satellite drifts and biases, while MWRI was not assimilated into ERA5. In addition, ERA5 has also assimilated a variety of observational data other than satellite observations. Therefore, due to the differences in the observation data and processing methods used, RSS and ERA5 products can be considered independent of our CDR. We have added the following sentence in the revised manuscript to address these points.

*Section 4.2, Lines 394–395*: "As a ground-based observation system, GNSS is completely independent of retrievals and offers a reliable local reference."

*Section 4.2, Lines 410–412*: "Since GTMBA is a moored in situ buoy network, it is fully independent of the satellite retrievals and serves as a reliable reference for evaluating SST trends at fixed ocean locations."

*Section 4.2, Lines 429–434*: "Although both RSS and ERA5 include AMSR-E and AMSR2 data, their processing chains are entirely independent of the RTV. As a peer satellite climate product, RSS uses a different retrieval algorithm and merges the measurements from additional sensors (e.g., SSM/I and TMI) but not MWRI (Mears et al., 2018). ERA5 also excludes MWRI and applies variational bias correction to remove satellite drifting biases, and assimilates observations into a model-based reanalysis system (Hersbach et al., 2020)."

3.  *Validation Data*. Different source of in-situ validation data is used. The authors should make clearer that these are of superior quality, independent, and consistent with each other.

**Reply: We thank the Anonymous Referee for raising this important point. The in-situ datasets used in our validation are widely recognized for their high quality, independence, and internal consistency. For CWV, the IGRA RAOB provide a robust reference, with a multi-stage quality assurance system that includes automated checks for persistence, climatological outliers, and vertical/temporal consistency, as described by Durre et al. (2008). Manual review–optimized thresholds and global applicability ensure minimal undetected errors (~1.1%) and no systematic biases. For SST, we use the GODAE SST from the FNMOC, which including data from ships, moored buoys, and drifting buoys. Buoy data are generally of higher quality, while ~8% of lower-quality ship data are excluded through decision-making quality control algorithms. Following Gentemann and Hilburn (2015), we only use in-situ measurements flagged as high quality, ensuring**

**that the validation results reflect reliable reference standards. The manuscript has been supplemented and revised accordingly.**

*Section 2.1, Lines 106–108*: "The RAOB data has undergone a multi-stage quality assurance process, including persistence checks, climatological outlier removal, and vertical/temporal consistency tests, which ensure internal coherence and minimize undetected errors to about 1.1% (Durre et al., 2008)."

*Section 2.1, Lines 118–120*: "The probability of gross error is used to ensure data quality and consistency, ensuring that only high-confidence data (defined as those with a probability less than 0.6 K) are used here (Gentemann and Hilburn, 2015)."

4. Especially, I am skeptical about using CMIP simulations as validation data. To underpin the value of using CMIP simulations in this article, the authors should clearly explain for what statistical metrics the CMIP simulations contribute to the validation of observations.

   The authors correctly indicate that the CWV and SST variables of CMIP relate to each other by design, via the Clausius Clapeyron law. This predetermined relationship makes that one cannot use both variables as independent sources of validation data. In the paper only one variable can be used for validation. Please comment on this.

**Reply: We thank the Anonymous Referee for this important and thoughtful comment. We fully agree that CMIP simulations should not be used as traditional reference data for validating the absolute accuracy of individual satellite retrievals. However, in our study, the use of CMIP simulations serves a different purpose— to evaluate the internal physical consistency of independently retrieved CWV and SST trends within the framework of thermodynamic constraints, as embodied in the Clausius-Clapeyron relationship. This aspect represents a distinctive component of our study. While most variables in CMIP cannot be directly compared with satellite products, Santer et al. (2021) has shown that the covariability between CWV and temperature products, is a suitable metric for such comparison. In particular, these inter-variable relationships can act as a bridge between model projections and satellite-observed responses to climate forcing.**

**In our analysis, we find that the CWV–SST trend ratio derived from our satellite retrievals aligns closely with that from the CMIP6 ensemble, reinforcing the notion that the CDR preserves large-scale thermodynamic coupling. This comparison does not treat CMIP as a "truth" dataset but instead uses it as a physically grounded expectation. When models and observations yield consistent coupling behavior, it suggests a successful mutual constraint: the models are credible, and the observations are physically self-consistent. Furthermore, Santer et al. (2021) emphasized that not all observational datasets are equally reliable when used to evaluate model behavior. In their findings, some CWV (e.g., RSS)**

**and SST products did not support the model-observed consistency in trends, raising the question of whether the discrepancy stemmed from errors in the models or the observations. At the time, this could not be conclusively determined. However, our result adds new evidence, showing that our independently retrieved CWV and SST trends are similar with CMIP expectations—suggesting that previous observational discrepancies may reflect observation limitations rather than model error.**

**We have revised the relevant section in the manuscript to make these points clearer, and now emphasize that the CMIP comparison serves as a mutual consistency check, not a direct validation of trend accuracy. We hope this clarification addresses the Anonymous Referee's concern.**

*Section 5, Lines 460–466*: "As mentioned earlier, the Clausius-Clapeyron relationship imposes a strong thermodynamic constraint on the covariance of CWV and temperature products under long-term warming. This physical linkage can be used to examine whether the joint behaviour of independently retrieved CWV and temperature variables conforms to expectations under radiative climate forcing. In fact, Santer et al. (2005, 2021) pointed out that this approach provides a meaningful way to evaluate observational datasets, with climate models offering physically grounded reference values. Following this rationale, we regard the trend ratios between CWV and SST (or TLT, TMT) as physically interpretable metrics that support mutual consistency checks between observations and model simulations."

*Section 5, Lines 512–516*: "This highlights the value of using climate covariance to evaluate the internal consistency of both observations and models. When retrieved satellite products reproduce the trend ratios expected from CMIP6 simulations, it increases confidence in their physical reliability. It also suggests that past disagreements between models and observations may have stemmed from biases in earlier datasets, rather than flaws in the models themselves."

5. *Uncertainty and error propagation.* The authors spend little words on the uncertainty of the CWV and SST products, and do not provide uncertainty bars in their figures. Literature describes approaches to estimate uncertainties of level 1 and level 2 satellite data, using metrological principles. Relevant work on this was done in the framework of the FIDUCEO project.

   FIDUCEO method paper:

   Giering, R.; Quast, R.; Mittaz, J.P.D.; Hunt, S.E.; Harris, P.M.; Woolliams, E.R.; Merchant, C.J. A Novel Framework to Harmonise Satellite Data Series for Climate Applications. Remote Sens. **2019**, *11*, 1002. https://doi.org/10.3390/rs11091002.

   FIDUCEO example paper:

Hans, I.; Burgdorf, M.; Buehler, S.A.; Prange, M.; Lang, T.; John, V.O. An Uncertainty Quantified Fundamental Climate Data Record for Microwave Humidity Sounders. Remote Sens. **2019**, *11*, 548. https://doi.org/10.3390/rs11050548.

More general descriptions for performing error propagation for Essential Climate Variables (ECVs) of the Global Climate Observing System, involving systematically tracking and quantifying uncertainties through all stages of the data processing (i.e., from raw observations to final climate data products), are described in the following papers:

Roebeling, R. A., S. Bojinski, P. Poli, V. O. John, and J. Schulz, 2025: On the Determination of GCOS ECV Product Requirements for Climate Applications. Bull. Amer. Meteor. Soc., **106**, E868–E893, https://doi.org/10.1175/BAMS-D-24-0123.1.

Merchant, C. J., Paul, F., Popp, T., Ablain, M., Bontemps, S., Defourny, P., Hollmann, R., Lavergne, T., Laeng, A., de Leeuw, G., Mittaz, J., Poulsen, C., Povey, A. C., Reuter, M., Sathyendranath, S., Sandven, S., Sofieva, V. F., and Wagner, W., 2017: Uncertainty information in climate data records from Earth observation, Earth Syst. Sci. Data, 9, 511–527, https://doi.org/10.5194/essd-9-511-2017.

**Reply: We thank the Anonymous Referee for this important and constructive comment. We have carefully reviewed the relevant literature, including the FIDUCEO framework and other uncertainty propagation methodologies for ECVs, and we find these works highly valuable and instructive. While our current study does not directly start from raw instrument counts, we have systematically quantified the uncertainties of the Level 1 FCDR TBs and the Level 2 retrieved CWV and SST products. For the Level 3 gridded products, where retrieval uncertainties are substantially reduced through spatial and temporal averaging, we follow the approach of Zou et al. (2023) to estimate the statistical uncertainties in long-term trend detection.**

**Specifically, for level 1 data, the FCDR relies on pre-processed TBs from three instruments, each with different measured uncertainties defined by the noise-equivalent temperature difference (NEΔT). For AMSR-E and AMSR2, we use resampled TBs computed using the Backus-Gilbert (BG) method (Kawanishi et al., 2003; Maeda et al., 2016), where the channel uncertainties are characterized as the product of the NEΔT and an empirically derived noise amplification factor. For MWRI, as discussed in Wu et al. (2020), the TBs used in this study have undergone a combination of sensor-specific calibration and inter-satellite harmonization techniques, including a nonlinear lookup-table (LUT)-based correction. This calibration process introduces strong nonlinearity and discontinuities, making it impossible to derive an analytical Jacobian matrix for traditional uncertainty propagation. Therefore, inspired by the methodologies**

described in the references suggested by the Anonymous Referee, we adopted a Monte Carlo approach: the uncorrected TBs were perturbed using their original NE$\Delta$T values, passed through the nonlinear correction process, and the standard deviation of the resulting ensemble was used as an estimate of the final uncertainty. This method effectively captures the impact of the nonlinear correction on uncertainty propagation. For AMSR-E, TBs use linear correction and the Jacobian matrix is close to 1. Therefore, the uncertainty of the corrected AMSR-E TBs is equivalent to the uncertainty of the resampled input. As a result, we obtained level 1 TB uncertainties for all three instruments.

For the Level 2 retrieval products, the total uncertainty arises from two main aspects: the propagation of TB uncertainties through the retrieval algorithm, and the intrinsic uncertainty of the algorithm itself. Given that the SST retrieval algorithm involves multiple threshold-based segmentations and nonlinear terms, it constitutes a nonlinear system. Therefore, we also use a Monte Carlo approach to propagate the input TB uncertainties through the retrieval process and estimate the associated retrieval uncertainty. In contrast, the CWV retrieval is expressed analytically as a logarithmic function of TBs, enabling the use of a Jacobian-based linear error propagation approach. We compared the CWV uncertainties computed by the analytical Jacobian method and the Monte Carlo method, and found that the difference between the two was less than 0.01 mm, indicating that the Monte Carlo approach is sufficiently accurate for both variables. The algorithmic uncertainty is further estimated by comparing the retrieved CWV and SST values against their "true" values in a large set of training data simulated using a microwave radiative transfer model. Based on the statistical differences between retrievals and true values, the standard algorithmic uncertainties were found to be 0.51 mm for CWV and 0.20 K for SST, both lower than the uncertainties reported by Wentz (2000). This improvement is attributed to the inclusion of the 10 GHz channel in the CWV retrieval and the use of second-order term in localized algorithm in the SST retrieval. Finally, the total Level 2 uncertainty is calculated by combining the retrieval-propagated uncertainty and the algorithmic uncertainty using standard uncertainty propagation formulas.

For Level 3 gridded products, retrieval uncertainties are further reduced through spatial and temporal averaging. Following the classification framework proposed by Zou et al. (2023), trend uncertainties in satellite-derived climate data records can be categorized into three components: structural, internal, and statistical uncertainty. Structural uncertainty arises from differences in instrument design, calibration approaches, and retrieval algorithms across independently generated datasets. Internal uncertainty refers to errors introduced within a specific data product, including those from calibration adjustments, diurnal drift corrections, sampling strategies, and uncertainty propagation. Statistical uncertainty, which becomes the dominant source at Level 3, is linked to the limited temporal coverage and internal variability of the climate system. This study primarily focuses on the statistical component, as it directly affects the robustness of long-term trend

estimates—particularly in the presence of strong low-frequency variability such as ENSO or volcanic signals. As highlighted by Santer et al. (2008), ignoring temporal autocorrelation in monthly anomalies can significantly underestimate trend uncertainty. To account for this, we follow their method by adjusting the effective sample size using the lagged autocorrelation coefficient, thereby obtaining more reliable standard errors and confidence intervals for linear trends. Specifically, we added the following to our manuscript.

*Section 3, Lines 338–345*: "Uncertainties in our dataset were systematically assessed across level-1 to level-3. Table 5 gives uncertainty estimates for level-1 and level-2. For level-1, the FCDR relies on pre-processed TBs from AMSR-E, AMSR2, and FY-3 MWRI. Channel-specific TB uncertainties for AMSR-E and AMSR2 are defined by the noise-equivalent temperature difference (NEΔT) and adjusted by noise amplification factors derived from the Backus–Gilbert (BG) resampling method (Kawanishi et al., 2003; Maeda et al., 2016). For MWRI, the complex recalibration introduces strong nonlinearity (Wu et al., 2020), for which analytical Jacobians are not available for uncertainty propagation. Thus, a Monte Carlo method (Merchant et al., 2017; Roebeling et al., 2025) was used to estimate the uncertainty after correction by perturbing input TBs with NEΔT and calculating the standard deviation of the corrected outputs."

*Section 3, Lines 348–355*: "For level-2, total retrieval uncertainty includes both the propagation of TB uncertainty and the intrinsic retrieval algorithm error. The CWV algorithm allows both Jacobian-based and Monte Carlo-based propagating uncertainty estimation (Giering et al., 2019; Hans et al., 2019), which yield similar results (<0.01 mm difference). The SST algorithm involves localized approach, requiring Monte Carlo simulation. Algorithmic uncertainties were further estimated by comparing retrievals with simulated truth from microwave radiative transfer model, yielding standard uncertainties of 0.51 mm for CWV and 0.20 K for SST. These values are lower than those reported by Wentz (2000), due to enhanced channel selection and algorithm design. Final level-2 uncertainties (Table 5) were obtained by combining both components using propagation of uncertainty (Giering et al., 2019; Hans et al., 2019)."

*Section 3, Lines 356–367*: "For Level-3 gridded products, retrieval uncertainties are substantially reduced through spatial and temporal averaging. In this study, we focus primarily on the statistical uncertainty (SU) associated with trend estimation, which is especially relevant given the limited record length and the presence of low-frequency climate variability (Zou et al., 2023). Following the approach of Santer et al. (2008), we account for temporal autocorrelation in the anomaly time series by adjusting the effective sample size using the lagged autocorrelation coefficient. This correction ensures that the derived confidence intervals for linear trends are not underestimated, providing a more reliable assessment of long-term climate signals. The SU estimates of the trends are expressed as 95% confidence level, is computed as:

$$SU = 1.96 \cdot SE \cdot \sqrt{\frac{1+r}{1-r}}, \tag{9}$$

where SE is the uncorrected standard error from ordinary least squares regression, and *r* is the lagged autocorrelation coefficient. Unless otherwise specified, *r* refers to the correlation between values that are 1 time period apart (lag-1 autocorrelation) throughout this study. The level-3 trend uncertainty is not included in Table 5, but is reported in all trend analysis in the relevant comparison figures below."

**Table 5. Uncertainty Estimates for level-1 TBs and level-2 retrievals.**

| Product Level | Channel / Parameter | | AMSR-E | MWRI | AMSR2 |
|---|---|---|---|---|---|
| level-1 | TB uncertainty (K) | 10V | 0.29 | 0.26 | 0.28 |
| | | 10H | 0.29 | 0.25 | 0.28 |
| | | 18H | 0.13 | 0.18 | 0.18 |
| | | 23V | 0.12 | 0.18 | 0.15 |
| | | 36H | 0.12 | 0.19 | 0.15 |
| level-2 | Retrieval uncertainty | CWV (mm) | 0.54 | 0.55 | 0.55 |
| | | SST (K) | 0.55 | 0.50 | 0.53 |

6. *Climate data records temporal coverage*. Climate studies often ask for data that covers the standard 30-year reference period defined by WMO, eg 1991-2020 (current reference period) or 2000-2030 (next reference period). Are there any plans to expand further back in time, e.g., back to 1997 by using TMI data, to cover the next reference period?

**Reply: We thank the Anonymous Referee for raising this important point. We fully agree that extending the temporal coverage of the climate data record (CDR) to meet the WMO-defined 30-year reference period is essential for long-term climate applications. To this end, our group are actively planning both backward and forward extensions of the current CDR. On the backward side, we are working to integrate microwave imager observations from legacy instruments, including SSMI (from 1987) and TMI (from 1997), which will allow us to extend the CWV record back to 1987 and the SST record back to 1997, respectively. These extensions are technically feasible given the similarity in channel configurations and the availability of inter-sensor calibration strategies already developed in previous studies. On the forward side, we are continuing the CDR using AMSR2 data beyond 2022, and incorporating observations from the Global Precipitation Measurement (GPM) Microwave Imager (GMI), the Special Sensor Microwave Imager/Sounder (SSMIS), and other new-generation sensors such as MWRI onboard subsequent FY-3 series satellites. With these planned extensions, the final dataset will span more than 30 years, fully satisfying the WMO criteria for reference climatology. We have added a statement regarding these extension plans in the revised manuscript.**

*Section 7, Lines 545–549*: "In particular, we plan to extend the CWV record back to 1987 using SSM/I and the SST record back to 1997 using TMI, respectively. Meanwhile, the forward extension will include ongoing AMSR2 observations and data

from next-generation MWRI instruments onboard subsequent FY-3 series satellites. This will enable coverage of the standard 30-year reference periods defined by the World Meteorological Organization (WMO), ensuring that the dataset remains comprehensive."

**Minor Criticisms**

7. Section 3: CDR Data: Add a table listing what data were used to construct the CDR, ie, name instrument the start date, end date, and discuss what steps were taken to harmonize and homogenise (see definitions https://research.reading.ac.uk/fiduceo/glossary/) these data over the entire CDR period, so as to use them for trend analysis.

**Reply: We thank the Anonymous Referee for this important comment. Wu et al. (2020) have conducted extensive and systematic recalibration work in the preliminary stage to support the construction of the FCDR. Specifically, these included correction of time-dependent calibration drifts using the vicarious cold reference method, removal of geolocation errors via the coastline inflection method, inter-sensor calibration through the double-difference (DD) method using AMSR2 as the reference, and a principal component analysis (PCA) approach to reduce hardware-induced biases over land. After these recalibrations, we believe that the FCDR TB has sufficient harmonization and homogenization. The biases of the matching pairs of MWRI and the other two instruments are close to 0 K, and the STDs of all used channels are all less than 0.7 K. These calibration procedures are not detailed in this paper. We have supplemented the instrument list and the constructing steps of the CDR in the manuscript.**

*Section 3, Lines 237–241*: "Table 3 summarizes the instruments and their time coverage used to construct the current CDR. To ensure consistency and long-term stability of the retrieved climate variables, a series of processing steps and adjustments were implemented. These steps include: i) removing precipitation and sea ice pixels, ii) correcting for diurnal drifts in MWRI, iii) retrieving CWV and SST, iv) spatially gridding the data onto a global $0.25° \times 0.25°$ grid, v) merging of multi-sensor anomalies into a continuous time series, and vi) comprehensive uncertainty quantification at each processing level. Details of each step are provided below."

Table 3. Satellite instruments contributing to the CDR.

| Satellite | Instrument | Start date | End date |
|-----------|-----------|-----------|----------|
| Aqua | AMSR-E | 2002-06 | 2011-06 |
| FY3B | MWRI | 2011-03 | 2015-04 |
| GCOM-W1 | AMSR2 | 2012-09 | 2022-05 |

8. *Page 9, line 232*: replace "thereby eliminated" by "thereby empirically correcting".

**Reply: Done!**

9. *Equation (1), (4), (5)*: For consistency with the other notations, can you replace

    $D(X, m, L)$, $TB(X, t, m, L)$, and $TB'(X, t, m, L)$

    with

    $D_{MWRI}(X, m, L)$, $TB_{MWRI}(X, t, m, L)$, and $TB'_{MWRI}(X, t, m, L)$

**Reply: Done!**

10. *Figure 3*: is the blue line in this figure not the difference of the two instruments before correction, thus:

    $TB_{MWRI}(X, t, m) - TB_{AMSR}(X, t, m)$

    Instead of

    $\Delta TB' = TB'(X, t, m) - TB_{AMSR}(X, t, m)$

    This is what I expect, because then the figure demonstrated that the difference between the instruments before corrections resemble those of the diurnal anomalies and thus seems to prove that a correction is needed. Please explain.

**Reply: Done! The Anonymous Referee is absolutely correct — the blue line represents the difference between unadjusted MWRI and AMSR TBs, thereby justifying the need for calibration correction. We also added the phrase "before diurnal drift adjustment" in the figure caption.**

**Figure 3: Inter-satellite differences ($TB_{MWRI}(X, t, m) - TB_{AMSR}(X, t, m)$) before diurnal drift adjustment (blue lines) and diurnal anomaly differences ($D_{MWRI}(X, m, L)$, red lines) derived from the semi-physical model over the global ocean.**

11. *Figure 6 (and similar figures later)*: Indicate in the caption that the numbers given in the legend for each validation result represent BIAS and RMSE.

**Reply: Thanks for the suggestion. The numbers shown in the legend actually represent the linear trend and its standard error. We have now clearly labeled these values in the figure legend, and also clarified in the caption of Figure 6, where this notation first appears. For example, the revised figure 6 are provided below.**

[Figure]

**Figure 6: Monthly anomaly time series for CWV_RTV, GNSS, ERA5, and RSS over the three GNSS sites. The legend in each panel includes statistical metrics for trend comparison. The black text indicates the linear trend of the RTV anomaly (black line) along with its SU. Colored texts represent the trend values, SUs, and RMSEs of the anomaly differences between RTV and each of the validation datasets (GNSS, ERA5, and RSS). As the anomaly differences are computed over identical time periods, the mean bias is inherently zero and thus not shown. The lag-3 autocorrelation is used for the BRMU station in a).**

12. *Figure 8 & 9*: The compared datasets are partly based on the same observational datasets (AMSR-E and AMSR2) and thus cannot be considered independent of each other. I realize that complete independence is difficult to achieve, still the authors discuss and provide evidence of the degree of independence of the compared datasets. This is especially important in determining the climatological significance of jumps in the time series.

**Reply: We thank the Anonymous Referee for this thoughtful and important comment. As noted in our Response to Comment #2, we have carefully discussed the independence between our CDR and the reference datasets used in Figures 8 and 9. In particular, for ERA5 and RSS, while they incorporate AMSR-E and AMSR2 data, their processing chains, algorithms, and sensor combinations differ significantly from ours (see Comment #2). Given the very different processing schemes between each product and our CDR, the similarity observed in both short-term variability and long-term trends is notable. This consistency suggests that all datasets capture robust climate signals, and can serve as a valid reference for climate validation. Corresponding clarifications have been added in the revised manuscript.**

*Section 4.2, Lines 433*: "These differences make them suitable, independent references for evaluating variability and trends."

13. *Figure 8 & 9*: With reference to my above point, there is a clear jump in values between the period 2002-2012 and 2016-2022. This jump seems rather to be related to a change in instrument than to a change in climate. Please comment.

**Reply: We appreciate the Anonymous Referee's observation. In our inter-sensor recalibration process, special attention was given to remove instrument jump while preserve physically meaningful climate signals. The apparent jumps in 2010 and 2016 coincide with well-documented El Niño events, which caused abrupt increases in global CWV and SST. These climate shifts are supported by the ENSO index (see: https://climatedataguide.ucar.edu/climate-data/nino-sst-indices-nino-12-3-34-4-oni-and-tni). Importantly, the OISST product, which is fully independent of our CDR in both observation and processing, also captures similar jumps during the same periods. This supports the interpretation that the changes observed between 2002–2012 and 2016–2022 are primarily climatic, rather than artifacts of instrument transition. We have added a clarification on this point in the revised manuscript.**

*Section 4.2, Lines 433–435*: "Although independent, including the fully independent OSSIT, all datasets exhibited a shift from 2010 to 2016 (Fig. 8). This most likely suggests that this shift is a true climate shift."

14. *Line 375*: The statement "Different datasets show overall similar trend patterns for both CWV and SST." is very qualitative. Can you provide some statistics to make it more quantitative?

**Reply: We agree with the Anonymous Referee that this sentence was overly qualitative. In response, we have added spatial correlation coefficients to quantify the similarity of trend patterns across datasets. The spatial correlation between CWV$_{RTV}$ and CWV$_{ERA}$ reaches 0.92, indicating high consistency. The slightly lower CC of 0.78 involving RSS CWV is due to differences in spatial resolution, as the comparison was performed on matched grid points only. The revised sentence now reads:**

*Section 4.2, Lines 448–449*: "Different datasets show overall similar trend patterns for both CWV and SST, with spatial CCs exceeding 0.78 and 0.85, respectively."

15. *Figure 11*: Why are there more crosses for the CMIP trends. Are these different CMIP scenarios?

**Reply: We thank the Anonymous Referee for this question. The crosses shown for CMIP in Figure 11 represent individual model simulations. Specifically, we selected 28 CMIP6 models, each using a concatenation of the historical experiment (up to 2014) and the SSP5-8.5 scenario (from 2015 onward). This high-emission scenario was chosen to reflect a strong forced climate signal over the analysis period (2003–2020). The large spread of trend estimates in either SST or CWV represents model uncertainties in simulating climate trends. However, our**

**primary focus is not on absolute trends, but rather on the ratios between CWV and temperature trends (e.g., $R_{\{CWV/SST\}}$, $R_{\{CWV/TLT\}}$), which are considered to be more physically constrained and thus more appropriate for comparing observations with models. These ratios serve as mutual benchmarks and have been proposed in prior studies (e.g., Santer et al., 2021) as useful indicators of physical consistency across datasets.**

16. *Line 452 Conclusions*: Is the statement "The most encouraging result is that the covariance between our retrieved CWV and SST over the tropical oceans is close to the expectations from CMIP6 model simulations." true?

    CMIP is an ensemble of model simulations, matching with CMIP does not say much about the quality of the observational data. May be one could reason the other way around and write that it is encouraging that the CMIP simulations seem to be able to reproduce the observed relationships. This would, however, be a statement about the quality of CMIP and not about the quality of the CWV and SST observations! Please comment.

**Reply: Please see our Reply on Comment #4.**

**Overall Recommendation**

17. Provide a pointwise description of what validation metric is evaluated with what reference data.

**Reply: We thank the Anonymous Referee for this helpful suggestion. To improve the clarity and traceability of our validation framework, we have added Section 2.8 in the revised manuscript, which provides a structured summary of each reference dataset, its degree of independence, and the specific validation metrics used. This information is also concisely presented in Table 2.**

*Section 2.8, Line 202*: "2.8 Validation Framework of Reference Datasets"

*Section 2.8, Lines 203–207*: "To ensure a structured evaluation of the CDR quality, we systematically validated the retrieved CWV and SST products across multiple dimensions: retrieval accuracy, local and global long-term trends, and climate trend covariability. The various datasets mentioned above differ in spatial coverage, temporal extent, measurement principles, and are therefore suitable for different validation tasks. Table 2 summarizes the validation objectives, evaluation metrics, corresponding reference datasets, and their independence from the retrievals."

**Table 2. Reference datasets used in validation. (RMSE: Root mean square error; CC: Correlation coefficient)**

| Validation Objectives | Variable | Validation Metric | Reference Dataset | Dependency |
|---|---|---|---|---|
| Retrieval accuracy | CWV | Bias, RMSE, CC | RAOB | Fully independent |
| | SST | Bias, RMSE, CC | GODAE | Fully independent |
| Regional variability and trend | CWV | Trend, CC | GNSS | Fully independent |
| | SST | Trend, CC | GTMBA | Fully independent |
| Global variability and trend | CWV | Trend, CC | RSS | AMSR-E and AMSR2 observations included |
| | SST | Trend, CC | OISST | Fully independent |
| | CWV and SST | Trend, CC | ERA5 | AMSR-E and AMSR2 observations included |
| | CWV and SST | Trend, CC | RSS-CDR | AMSR-E and AMSR2 observations included |
| Climate trend covariability | CWV, SST, TLT and TMT | $R_{\{CWV/SST\}}$, $R_{\{CWV/TLT\}}$ and $R_{\{CWV/TMT\}}$ | CMIP6 | Fully independent |
| | TLT and TMT | $R_{\{CWV/TLT\}}$ and $R_{\{CWV/TMT\}}$ | STAR | Fully independent |
| | TLT and TMT | $R_{\{CWV/TLT\}}$ and $R_{\{CWV/TMT\}}$ | RSS | Fully independent |
| | TLT and TMT | $R_{\{CWV/TLT\}}$ and $R_{\{CWV/TMT\}}$ | UAH | Fully independent |

18. Explain to what extent the reference data are independent of, and superior to, the CDR.

**Reply:Please see our Reply on Comment #2 and Comment #17.**

19. Explain to what extent the CWV and SST products are independent of each other.

**Reply:Please see our Reply on Comment #1.**

20. Explain how uncertainties from input data and processing steps propagate into the CDR product.

**Reply:Please see our Reply on Comment #5.**

21. Discuss, and if possible assess, the role of validation uncertainties arising from collocation, synchronization, and representation differences between the reference data and the CDR.

**Reply:** We thank the Anonymous Referee for raising this important point. As discussed in Giering et al. (2019) and Hans et al. (2019), validation uncertainty arises not only from the intrinsic errors of each dataset, but also from mismatches in collocation, synchronization, and physical representation between the CDR and the reference data. According to Zou et al. (2006), spatial collocation errors—especially in variables with large horizontal gradients—can increase standard deviation, but tend to have symmetric distribution that reduces their impact on mean biases. In this study, we adopt matching criteria that balance accuracy and sample size. For RAOB, a typical radiosonde ascent takes about 2 hours and may drift more than 100 km laterally (Ingleby et al., 2016). Therefore, a collocation window of 60 km and 3 hours is considered appropriate. For GODAE SST, we use a tighter spatial threshold of $0.1°$ and temporal threshold of 6 minutes, as its field of view after resampling (over 20 km) is sufficiently large to absorb small-scale mismatch errors.

Regarding representation differences, the CWV retrieved from satellites includes water vapor from sea level to the top of the atmosphere, while RAOB profiles may miss low-level vapor depending on the station elevation. Similarly, satellite-based microwave SST represents the sub-skin temperature at millimetre depths, while in situ measurements reflect temperature at 1 m depth, leading to possible systematic offsets (e.g., https://cersat.ifremer.fr/Thematics/Sea-Surface-Temperature/Defining-sea-surface-temperature/). These factors are inherent in satellite–in situ comparisons and are taken into account when interpreting the validation results. We have made revisions in the manuscript.

*Section 4.1, Lines 378–381*: "It is worth noting that some of the residual variability may originate from collocation and representation mismatches (Zou et al., 2006; Giering et al., 2019; Hans et al., 2019). Radiosonde ascents typically take up to two hours and can drift over 100 km (Ingleby et al., 2016). Moreover, satellite-derived CWV includes the full atmospheric column, whereas RAOB may miss water vapor near the surface depending on station elevation (Buehler et al., 2012)."

*Section 4.1, Lines 384–387*: "The observed differences can be partially attributed to representation differences between microwave and in situ measurements. While satellite sensors retrieve sub-skin temperature at millimeter depth, GODAE integrates in situ measurements from ships and buoys, primarily at a depth of 1 meter (Huang et al., 2020)."

**Reference**

Buehler, S. A., Östman, S., Melsheimer, C., Holl, G., Eliasson, S., John, V. O., Blumenstock, T., Hase, F., Elgered, G., Raffalski, U., Nasuno, T., Satoh, M., Milz, M., and Mendrok, J.: A multi-instrument comparison of integrated water vapour

measurements at a high latitude site, Atmos. Chem. Phys., 12, 10925-10943, https://doi.org/10.5194/acp-12-10925-2012, 2012.

Durre, I., Vose, R. S., and Wuertz, D. B.: Robust automated quality assurance of radiosonde temperatures, J. Appl. Meteorol. Clim., 47, 2081-2095, https://doi.org/10.1175/2008jamc1809.1, 2008.

Giering, R., Quast, R., Mittaz, J. P. D., Hunt, S. E., Harris, P. M., Woolliams, E. R., and Merchant, C. J.: A novel framework to harmonise satellite data series for climate applications, Remote Sens.-Basel, 11, ARTN 1002, https://doi.org/10.3390/rs11091002, 2019.

Hans, I., Burgdorf, M., Buehler, S. A., Prange, M., Lang, T., and John, V. O.: An uncertainty quantified fundamental climate data record for microwave humidity sounders, Remote Sens.-Basel, 11, ARTN 548, https://doi.org/10.3390/rs11050548, 2019.

Ingleby, B., Pauley, P., Kats, A., Ator, J., Keyser, D., Doerenbecher, A., Fucile, E., Hasegawa, J., Toyoda, E., Kleinert, T., Qu, W. Q., James, J. S., Tennant, W., and Weedon, R.: Progress toward high-resolution, real-time radiosonde reports, B. Am. Meteorol. Soc., 97, 2149-2161, https://doi.org/10.1175/Bams-D-15-00169.1, 2016.

Merchant, C. J., Paul, F., Popp, T., Ablain, M., Bontemps, S., Defourny, P., Hollmann, R., Lavergne, T., Laeng, A., de Leeuw, G., Mittaz, J., Poulsen, C., Povey, A. C., Reuter, M., Sathyendranath, S., Sandven, S., Sofieva, V. F., and Wagner, W.: Uncertainty information in climate data records from earth observation, Earth Syst. Sci. Data, 9, 511-527, https://doi.org/10.5194/essd-9-511-2017, 2017.

Roebeling, R. A., Bojinski, S., Poli, P., John, V. O., and Schulz, J.: On the determination of GCOS ECV product requirements for climate applications, B. Am. Meteorol. Soc., 106, E868-E893, https://doi.org/10.1175/Bams-D-24-0123.1, 2025.

Santer, B. D., Thorne, P. W., Haimberger, L., Taylor, K. E., Wigley, T. M. L., Lanzante, J. R., Solomon, S., Free, M., Gleckler, P. J., Jones, P. D., Karl, T. R., Klein, S. A., Mears, C., Nychka, D., Schmidt, G. A., Sherwood, S. C., and Wentz, F. J.: Consistency of modelled and observed temperature trends in the tropical troposphere, Int. J. Climatol., 28, 1703-1722, https://doi.org/10.1002/joc.1756, 2008.

Zou, C. Z., Goldberg, M. D., Cheng, Z. H., Grody, N. C., Sullivan, J. T., Cao, C. Y., and Tarpley, D.: Recalibration of microwave sounding unit for climate studies using simultaneous nadir overpasses, J. Geophys. Res.-Atmos., 111, Artn D19114, https://doi.org/10.1029/2005jd006798, 2006.

Zou, C. Z., Xu, H., Hao, X., and Liu, Q.: Mid-Tropospheric layer temperature record derived from satellite microwave sounder observations with backward merging approach, J. Geophys. Res.-Atmos., 128, e2022JD037472, https://doi.org/10.1029/2022JD037472, 2023.